# A data-driven model for Fennoscandian wildfire danger

Sigrid Jørgensen Bakke[1], Niko Wanders[2], Karin van der Wiel[3], and Lena Merete Tallaksen[1]

[1]Department of Geosciences, University of Oslo, Norway
[2]Department of Physical Geography, Utrecht University, the Netherlands
[3]Royal Netherlands Meteorological Institute, the Netherlands

**Correspondence:** Sigrid J. Bakke (s.j.bakke@geo.uio.no)

**Abstract.** Wildfires are recurrent natural hazards that affect terrestrial ecosystems, the carbon cycle, climate and society. They are typically hard to predict, as their exact location and occurrence are driven by a variety of factors. Identifying a selection of dominant controls can ultimately improve predictions and projections of wildfires in both the current and a future climate. Data-driven models are suitable for identification of dominant factors of complex and partly unknown processes, and can both help improve process-based models and work as independent models. In this study, we applied a data-driven machine learning approach to identify dominant hydrometeorological factors determining fire occurrence over Fennoscandia, and produced spatiotemporally resolved fire danger probability maps. A random forest learner was applied to predict fire danger probabilities over space and time, using a monthly (2001–2019) satellite-based fire occurrence dataset at a 0.25° spatial grid as the target variable. The final data-driven model slightly outperformed the established Canadian fire weather index (FWI) used for comparison. Half of the 30 potential predictors included in the study were automatically selected for the model. Shallow volumetric soil water anomaly stood out as the dominant predictor, followed by predictors related to temperature and deep volumetric soil water. Using a local fire occurrence record for Norway as target data in a separate analysis, the test set performance increased considerably. This demonstrates the potential of developing reliable data-driven models for regions with a high quality fire occurrence record, and the limitation of using satellite-based fire occurrence data in regions subject to small fires not identified by satellites. We conclude that data-driven fire danger probability models are promising, both as a tool to identify the dominant predictors and for fire danger probability mapping. The derived relationships between wildfires and the selected predictors can further be used to assess potential changes in fire danger probability under different (future) climate scenarios.

## 1 Introduction

Boreal ecosystems, covering large parts of northern North America and Eurasia, comprise one of the world's most extensive biomes (Balshi et al., 2009). In the boreal region, which have the largest carbon stock of all major global forest biomes, fires are the major stand-renewing agent and play a major role in carbon storage and emissions (Bradshaw and Warkentin, 2015; Flannigan et al., 2009). Boreal fires emit up to 9% of global fire carbon emissions and 15% of global fire methane emissions annually (Van der Werf et al., 2017; Flannigan et al., 2009). The fire season length, fire frequency and burned area have increased in many parts of the boreal region, and these changes have been linked to climate change (Tomshin and Solovyev,

2021; Feurdean et al., 2020; Hanes et al., 2019; Balshi et al., 2009; Kasischke and Turetsky, 2006). Accordingly, improved knowledge about boreal fires and their occurrence is of high importance, both in the current and in future climate scenarios. Fires have been extensively studied in the boreal zones of North America and Russia (e.g. Tomshin and Solovyev, 2021; Hanes et al., 2019; Forkel et al., 2012; Balshi et al., 2009). However, to the best of our knowledge, fire studies within the European boreal zone are limited.

The European boreal zone covers Fennoscandia, a peninsula comprising Norway, Sweden, Finland, as well as the north-western part of Russia (Kola Peninsula and Karelia). Fennoscandia is known for its large spatial heterogeneity in hydroclimatological characteristics resulting from a high local variability in altitude, soil characteristics and moisture sources over short distances, to mention a few (Sømme, 1960). Similar to other boreal regions (Skinner et al., 2002), the fire season comprises the warm and dry period of the year, and is limited by snow during winter. Fires in Fennoscandia are normally small in size, and of shorter duration (more quickly distinguished), as compared to boreal regions in North-America and Siberia (Aalto and Venäläinen, 2021). However, there are recent examples of warm and dry summers where large areas burned, such as the record-breaking area burned in Sweden in 2018 (Krikken et al., 2021; San-Miguel-Ayanz et al., 2019). Such extreme fires pose the question of what to expect in the future.

Wildfires are recurrent natural hazards and an integral part of all major biomes (Keywood et al., 2013; Bowman et al., 2009). Wildfires both affect and are being affected by climate, emphasising the importance of incorporating fire activity when investigating the earth system. On both short and long time-scales, wildfires affect regional and global climate by changing the terrestrial ecosystem composition and functioning, surface energy fluxes, and the water and carbon cycle (Walker et al., 2019; López-Saldaña et al., 2015; Keywood et al., 2013; Flannigan et al., 2009). Wildfires are complex phenomena, driven by a combination of available biomass to burn, hydrometeorological conditions suitable for combusting and propagating the fire, and a source of ignition (Krawchuk and Moritz, 2011; Krawchuk et al., 2009). The hydrometeorological conditions are the most variable and largest driver of burned area (Jolly et al., 2015; Abatzoglou and Kolden, 2013; Littell et al., 2009; Flannigan et al., 2005; Bessie and Johnson, 1995), controlling whether or not an ignition leads to a fire. The hydrometeorological controls act on both seasonal to annual time scales, for example by controlling the presence of snow, and the moisture content of the soil and vegetation; and on short time scales, for example by concurrent hot, dry and windy weather. This makes wildfire occurrence a complex hazard that can be caused by a multitude of, typically statistically dependent, external drivers.

Research that explicitly takes into account information of observed fires are typically either large-scale assessments using satellite-based burned area products, or limited to regions where historical fire records are good and that recently have experienced devastating large wildfires (e.g. Kuhn-Régnier et al., 2021; Hanes et al., 2019; Kganyago and Shikwambana, 2020; Lizundia-Loiola et al., 2020; Turco et al., 2013; Andela et al., 2017; Aldersley et al., 2011; Kasischke and Turetsky, 2006). Because satellite-based fire information relies on reflectance changes from medium resolution sensors, such data can be very different compared to national historical fire records, which are typically of higher resolution. In particular, satellite-based burned area products suffer from a systematically underestimation of burnt area due to difficulties in detecting small fires (Padilla et al., 2015; Randerson et al., 2012). On the other hand, the availability and quality of national fire datasets varies among countries, whereas satellite products allow for consistent transboundary fire data.

Several fire characteristics are of interest, such as the fire regime, emissions, burned area, duration and feedbacks with vegetation. The characteristics all have in common that they rely on the fundamental question of when and where fires are likely to occur. Fire occurrence, or likelihood thereof (i.e. fire danger probability), can be used for monitoring, forecasting and projections, and has been investigated using three main approaches: fire weather indices, global fire models, and data-driven models, as further elaborated below.

The relation between hydrometeorology and wildfires have traditionally relied on established *fire weather indices* (e.g. Van Wagner et al., 1987; Bradshaw, 1984; Noble et al., 1980). These indices are used for fire danger mapping applicable for monitoring, forecasting and projections. Fire danger can be defined as the weather conditions that can trigger and sustain wildfires (Ranasinghe et al., 2021), and thus differs from (and is a prerequisite for) fire occurrence that additionally require an ignition. Fire weather indices are typically based on empirical and semi-physical equations relating weather observations to dryness of fuel, with the aim of determining the fire danger. Fire weather indices can also be calculated based on large-scale gridded reanalysis and climate model data (e.g. McElhinny et al., 2020), allowing for spatially continuous estimates. Such estimates are used for assessments of historical and future changes in fire danger (Sun et al., 2019; Abatzoglou et al., 2019; Jolly et al., 2015; Flannigan et al., 2013), whereas fire weather indices calculated using numerical weather forecast models are used for transnational fire monitoring and forecasting (San-Miguel-Ayanz et al., 2012). Despite their widespread use, fire weather indices are typically developed for specific countries or biomes, and are thus not necessarily well-suited for other regions and climates (Arpaci et al., 2013; Dowdy et al., 2009).

Over the last two decades, *global fire models* (fire-enabled Dynamic Global Vegetation Models; DGVMs) that can be coupled with climate models have been developed (Hantson et al., 2016). Most global fire models are process-based models that estimate fuel load and fuel moisture, based on fire weather indices, or other atmospheric and simulated moisture conditions. These estimates are subsequently combined with the probability of lightning and/or anthropogenic ignition, to determine whether a fire will occur in a grid cell. Lightning rates can be constant in time or based on statistics, whereas population density is typically used to estimate human ignition probabilities as well as human suppression of fires. An advantage of global fire models is that they allow for examining the feedbacks between fire, vegetation and climate (Hantson et al., 2016). However, this relies on how well the model is able to represent reality. Whereas the global fire models have shown to reproduce the seasonality in burned area well, they vary in their ability to represent the spatial pattern in burned area, and are generally unable to represent interannual variations (Hantson et al., 2020).

Finally, *data-driven* (statistical and machine learning) *models* have been developed to relate wildfires to environmental and meteorological conditions (e.g. Gudmundsson et al., 2014; Aldersley et al., 2011). Unlike process-based approaches (fire weather indices and global fire models) that are constructed using pre-defined equations to relate a set of drivers to the fire characteristic of interest; a data-driven model follows a bottom-up approach that starts by considering the fire characteristic explicitly, and relates that characteristic to the combination of drivers based on the data itself. Many of the data-driven model studies predict spatial patterns in fire occurrence or burned area (Krawchuk et al., 2009; Parisien and Moritz, 2009; Prasad et al., 2008), sometimes over a typical fire season (Bedia et al., 2015; Gudmundsson et al., 2014; Aldersley et al., 2011). A few studies account for year-to-year variability by predicting the annual burned area (Littell et al., 2009; Balshi et al., 2009).

Data-driven model studies accounting for both seasonal and inter-annual variability in fire occurrences are limited. Those that exist typically predict monthly global patterns in burned area using predictors from observational data (Forkel et al., 2017), DGVMs (Forkel et al., 2019) or a combination of observational and reanalysis data (Kuhn-Régnier et al., 2021).

Data-driven methods are restricted to regions and applications that have sufficient data to both train the models and validate their performance. Although there are multiple examples of the opposite, a data-driven model should always be evaluated using a (part of the) dataset not used in the construction of the model. This is important to avoid overfitting, i.e. to ensure the model is able to predict the system of interest and not only the data points it is trained on. Another challenge is the rare occurrence of forest fires implying a highly skewed dataset, an imbalance that needs to be accounted for in the training and evaluation of a model. In addition, a bottom-up approach is typically less straightforward in its data requirements and methodology as compared to the process-based approaches, because a bottom-up approach is not limited by the physical understanding of the system, and the amount of data and algorithms implemented are in principle unlimited.

Despite these challenges, bottom-up approaches are valuable as they facilitate an explicit link between climate science and societal or environmental impacts, as emphasised by the compound event framework among others (Van der Wiel et al., 2020; Zscheischler et al., 2018; Leonard et al., 2014). The approach allows for a high degree of flexibility in terms of potential drivers, target variables and models. This flexibility allows for both detailed regional investigations, and constructing models transferable to different (future) climate scenarios (e.g. Goulart et al., 2021). As opposed to fire weather indices and global fire models, data-driven models do not require a priori assumptions of the dominant mechanisms and physical processes controlling fire occurrences (except indirectly in the selection of potential predictors). This makes data-driven models suitable when the nature of the processes is complex and/or partly unknown, and when the emphasis is on accurate predictions or identification of dominant controlling mechanisms (Parisien and Moritz, 2009). Data-driven models can in this way both help improve process-based models, and work as independent models. A data-driven approach allows us to identify controlling factors for fire occurrences for specific regions. This is especially important for regions such as Fennoscandia, which possess highly variable hydroclimatological conditions, and where the current number of (satellite-detected) fires are relatively low as compared to other regions of the world.

In this study, we developed a temporally and spatially explicit data-driven machine learning model for Fennoscandia to reach two main objectives:

– Identify the dominant predictors of wildfires

– Construct monthly fire danger probability maps

A satellite-based burned area product was used to construct the target dataset of fire occurrences over the period 2001–2019 at a monthly time step. We chose a random forest (RF) algorithm to train the model and identify dominant predictors from a predefined set of 30 hydrometeorological and land cover based indices. To have trust in the model, it was tested on an independent dataset using the Area Under the Curve of the Receiver Operating Characteristic (ROC-AUC). In addition, we aim to answer the following research questions:

1. How well does the data-driven model perform as compared to the Canadian Fire Weather Index (FWI) that is developed for similar biomes and latitudes as Fennoscandia?

2. How well does the data-driven model perform when applied to an independent local (Norwegian) fire occurrence dataset?

3. Does the performance of a data-driven model improve when using a local fire occurrence dataset as target variable for training the model?

Finally, we performed two additional experiments to challenge the model choices made:

4. Does the data-driven model chosen outperform both a simpler machine learning algorithm (Decision Tree), as well as a more sophisticated (AdaBoost) machine learning algorithm?

5. What is the effect of not including a dynamical vegetation index?

The paper is structured as follows: Sect. 2 provides a detailed description of the data sources and methods applied. The results are presented in Sect. 3. A discussion is provided in Sect. 4, followed by concluding remarks given in Sect. 5. The Supplement comprises Fig. S1–S13.

## 2 Data and methods

A general outline of the data-driven approach for Fennoscandia is shown in Fig. 1. Data and pre-processing of the target variable (fire occurrence) and the potential predictors are summarised in Table 1 and described in Sect. 2.1-2.3. Section 2.4 describes the data and calculation of the Canadian Fire Weather Index (FWI), which was used as an alternative fire danger model. The data-driven model set-up and training is described in Sect. 2.5. Section 2.6 describes the selection and evaluation of the final data-driven model, including the predictor importance estimate, comparison of fire danger probability maps produced by the data-driven model and FWI, and model evaluation using the Norwegian fire occurrence dataset. Finally, Sect. 2.7 describes the additional experiments challenging the model choices made.

### 2.1 Data and pre-processing of the target variable

Two spatiotemporally varying datasets were used as target variables for the analysis, a main satellite-based fire occurrence dataset (Sect. 2.1.1), and an additional Norwegian fire occurrence dataset (Sect. 2.1.2. The two datasets are fundamentally different; whereas the satellite-based fire occurrence dataset only captures fires large enough to impact the reflectance captured by the satellite, the Norwegian fire occurrence dataset comprises all wildfire occurrences, whereof many cover a rather small area.

### 2.1.1 Satellite-based fire occurrence dataset

Data used for the target variable was the gridded fire burned area European Space Agency Climate Change Initiative (ESA-CCI) product version 5.1.1cds, downloaded from the Copernicus Data Store (ESA-CCI, 2020). This dataset holds the same

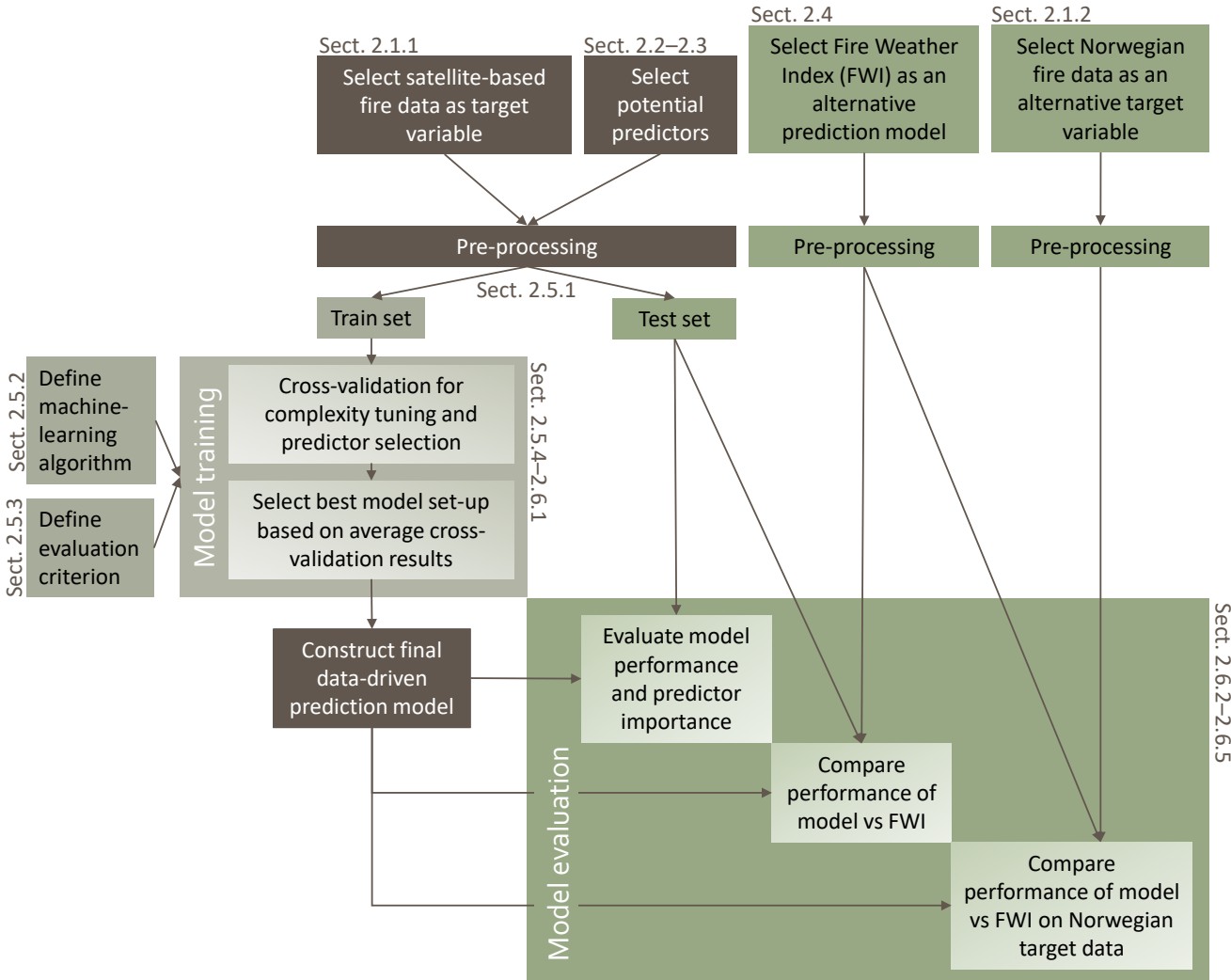

**Figure 1.** Flow chart of the data-driven approach for Fennoscandia. Sections given in the flow chart refer to sections in the text where detailed descriptions can be found. Grey, green and dark brown boxes represent steps relevant to model training, model evaluation and both, respectively.

information as the Fire ECV Climate Change Initiative Project (Fire CCI) burned area product version 5.1, and is based on the reflectance product of the Moderate Resolution Imaging Spectroradiometer (MODIS) sensor. The main reflectance data used

are daily surface reflectance information in the red and Near Infrared bands (more details found in Pettinari et al., 2019). Data uncertainties are related to a potential underestimation of the actual burned area due to cloud cover, haze or other low quality of the observations. The fire burned area dataset is available both as a 0.25° longitude/latitude regular grid product and as a pixel product of 250 m resolution. We chose to use the grid product to investigate if a data-driven model is applicable for use at

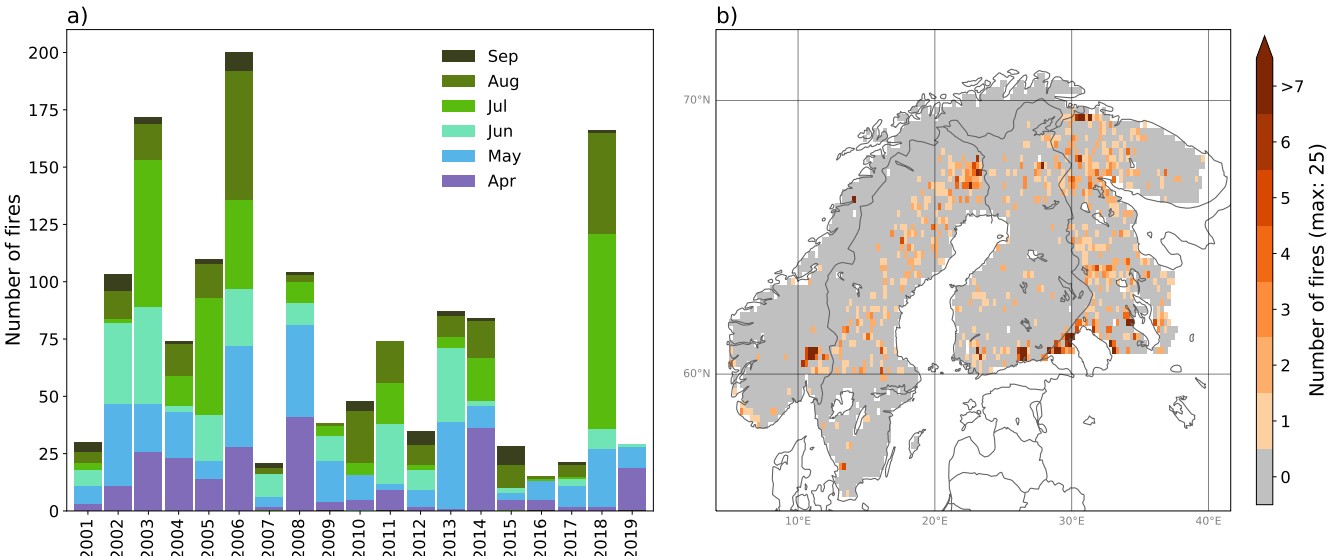

**Figure 2.** Distribution of fire occurrences in the satellite-based target variable over a) time (month and year) and b) space (0.25° resolution).

the spatial scale of the state of the art global climate models. Further, spatial dependency of fires (e.g. the same fire occurring in two or more cells) is reduced when using the coarser scale of the grid product as compared to the pixel product.

From the global fire burned area dataset, Fennoscandia was selected, and a two-class target variable of fire occurrence was constructed by classifying each data point as 1 (fire) if the burned area of the data point exceeded zero, and 0 (no-fire) otherwise. Thus, each data point (i.e. each grid cell for each time step) was independently considered as either a fire occurrence or not, and no merging was performed of fires that potentially extended multiple grid cells or months. The original monthly time step, period 2001–2019, and 0.25° longitude/latitude regular grid of the dataset were kept to have as many data points as possible. However, the months October to March were omitted each year, as they had less than 20 fire occurrences over the whole period and all grid cells, which was considered too few occurrences for the analysis. Figure 2 shows the distribution of the fire occurrences over time and space. There is an extreme imbalance between the two classes (fire and no-fire) in the target variable, with only 1439 of the 444030 data points (0.3%) classified as fire.

### 2.1.2 Norwegian fire occurrence dataset

A national record of historical wildfires in Norway from the Norwegian Directorate for Civil Protection (DSB, 2020) was used to evaluate the model prediction capability using a different target dataset. The Norwegian fire occurrence dataset covers point location and date of wildfires in Norway from 2016 to near-present. The dataset comprises all fires registered in grass, cultivated land, forests and uncultivated land, regardless of ignition source. The data is based on the fire and rescue service reporting system in Norway (brann- og redningstjenestens rapporteringssystem; BRIS). There is no lower limit of burned area in this dataset, as it is based on fire responses of the fire department. The point locations in the dataset are the fire response

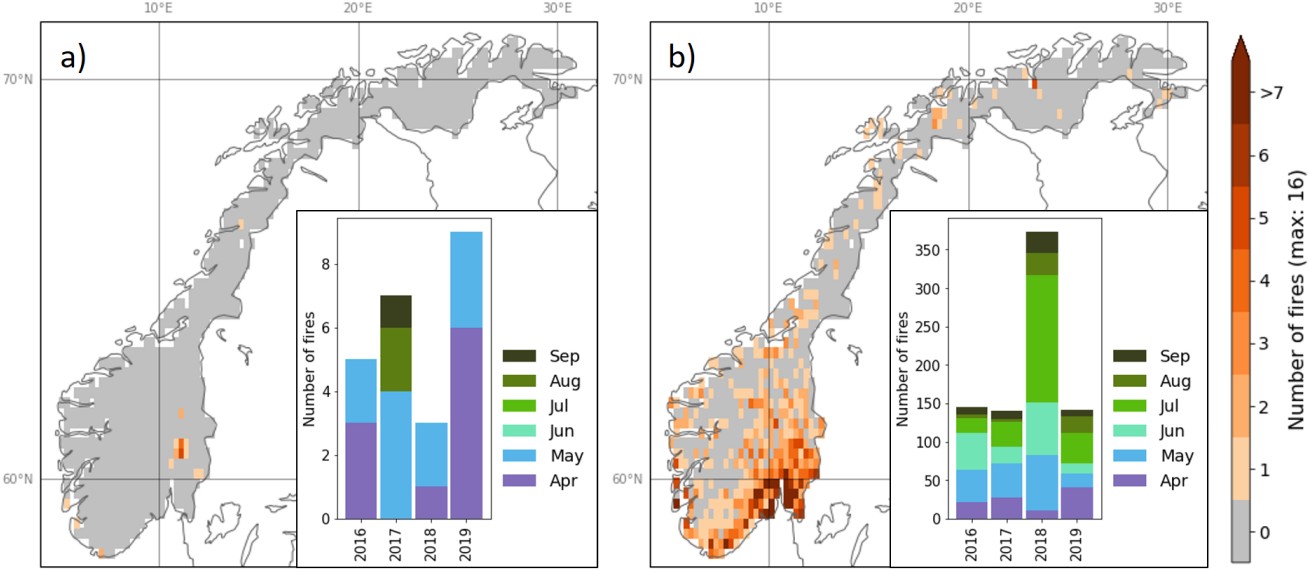

**Figure 3.** Distribution of fire occurrences in Norway using a) the satellite-based dataset and b) the Norwegian dataset. The figures show the spatial distribution over Norway (map; 0.25° resolution), and the temporal distribution over the period 2016–2019 (bar plot), as defined by the spatial and temporal domain of the Norwegian fire occurrence dataset. Note the different y-axes of the two bar plots.

attendance locations. Although these locations may not overlap with the locations where the fire started, we consider this uncertainty of minor importance at the 0.25° spatial grid applied in the study.

Each fire was assigned to the nearest grid cell using the same spatial grid as the satellite-based fire occurrence dataset, and
185 to the month of occurrence. All data points were classified as 1 (fire) if one or more fires occurred, and 0 (no-fire) otherwise. Because multiple fires can occur within the same month and grid cell, the total number of fire occurrences is lower for this variable than the original fire record data. Data covering the same season and period as the satellite-based fire occurrence dataset were selected, i.e. April–September 2016–2019.

Figure 3 shows the differences in the spatial and temporal distribution of fire data points (i.e. grid cells with recorded fire
occurrence at a given time step) in the Norwegian fire occurrence dataset and the satellite-based fire occurrence dataset for the spatial and temporal domain of the Norwegian fire occurrence dataset. There are substantial differences between the two datasets, mainly arising from the lack of small fires in the satellite-based fire occurrence dataset. The Norwegian dataset has in total 800 fire data points, as opposed to 24 in the satellite-based dataset. Whereas the three months of the highest number of fire data points in the Norwegian dataset are May–July, no fire data points exist for June and July in the satellite-based dataset.
The unusual high number of fires in 2018 is not reflected in the satellite-based dataset. Finally, the Norwegian dataset show a higher spread in fire occurrence across Norway as compared to the satellite-based dataset.

## 2.2 Criteria for potential predictors

The selection of potential predictors was governed by three criteria: available in most climate models, transferable to different climate scenarios, and compatible with the spatiotemporal resolution and domain of the target data. In addition, the data should be of high quality and the predictors should have a physical interpretation.

By limiting the selection of potential predictors to the two first criteria, we allow for investigations of fire danger probability as modelled by the data-driven model for different potential future climate scenarios. Due to the lack of human influence being represented in many climate models, predictors such as human infrastructure, settlement, and ignition sources were excluded from the analysis. Also lightning was excluded, partly due to the lack of such information in most climate models, partly due to the limited information such data can provide at a monthly 0.25° resolution, and partly due to the inconsistency in having only one type of ignition information. Other categories of potential predictors that were excluded include predictors that indirectly hold hydrometeorological information, such as month number, as well as dynamic vegetation related predictors, such as greenness indices. Dynamic vegetation related predictors were excluded because most climate model outputs are not based on runs for which the climate model is coupled with a Dynamic Global Vegetation Model (DGVM), but rather use prescribed vegetation cycles.

The third criterion ensured compatibility of all data used in the analysis. In order to carry out the data-driven approach, all potential predictors need to have the same spatiotemporal domain and resolution as used for the satellite-based target variable. Thus, all potential predictors were spatially constrained to Fennoscandia using a spatial resolution of 0.25°, and consisted of monthly values from April to September over the period 2001–2019. After the preparation of the potential predictors, the spatial domain was further limited to grid cells for which all data (including the target variable) had values for all time steps, and to grid cells where the fraction of burnable area (fraction_burnable) was greater than zero.

## 2.3 Data and pre-processing of potential predictors

The final selection comprised 30 potential predictors, several of which were highly correlated (Fig. 4). The derivation of each potential predictor is described in the following sections, ordered by categories as given in Table 1 (first column).

### 2.3.1 Precipitation, temperature and meteorological drought indices

The precipitation and temperature predictors were calculated based on ensemble mean daily precipitation totals (rr), and daily minimum, maximum and mean air temperature (tn, tx and tg, respectively) at 0.25° longitude/latitude from E-OBS version 23.1e (Cornes et al., 2018). E-OBS is a European dataset based on the European Climate Assessment and Dataset station information (ECA&D), covering the period 1950 until near-present.

From the daily values, monthly precipitation sums (rr_sum) and monthly means of tn, tx and tg (tn_mean, tx_mean and tg_mean, respectively) were calculated. The corresponding normalised anomalies (rr_sum_anomaly, tn_mean_anomaly, tx_mean_anomaly and tg_mean_anomaly) were calculated by subtracting the 1991–2020 mean and dividing by the 1991–2020 standard devia-

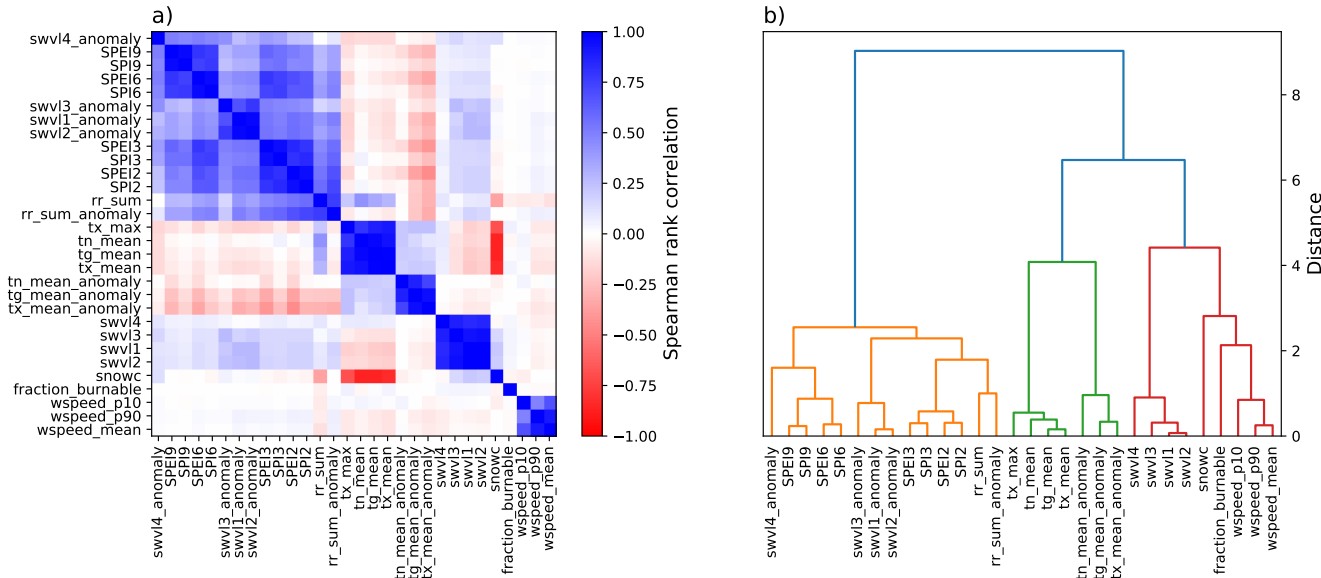

**Figure 4.** Statistical dependency of the potential predictors (abbreviations according to Table 1) using the full dataset: a) Spearman rank correlation and b) hierarchical clustering using Ward's Method (Ward, 1963).

tion of each month separately. The reference period 1991–2020 was chosen to follow the most recent normal period (WMO guidelines). An additional index, tx_max, was constructed by extracting the maximum daily tx for each month.

Two meteorological drought indices were included; the Standardized Precipitation Index (SPI; Guttman, 1999; McKee et al., 1993) and Standardized Precipitation-Evapotranspiration Index (SPEI; Beguería et al., 2014; Vicente-Serrano et al., 2010), each calculated for a 2, 3, 6 and 9 month accumulation period. The accumulation period is added to the abbreviation, e.g. SPI3 (representing a 3-month accumulation period). Both SPI and SPEI are indicators of the dryness/wetness of a site as compared to normal, and can be compared across locations with different climatology and highly non-normal precipitation distributions

(Stagge et al., 2014). Whereas SPI is an estimate of the precipitation anomaly, SPEI estimates the anomaly in the climatic water balance defined as precipitation minus potential evapotranspiration (PET). We used the Hargreaves equation (Hargreaves and Samani, 1985) to estimate PET, following the recommendation by Stagge et al. (2014). The Hargreaves equation estimates daily PET based on each day's tg, a proxy for net radiation (tx minus tn), and an estimate of extraterrestrial radiation based on the grid latitude and day of the year. To compute the SPI (SPEI), the precipitation (precipitation minus PET) for a given

accumulation period during the reference period is fitted to a parametric distribution, then non-exceedance probabilities from the distribution is transformed to the standard normal distribution, and finally the standard normal distribution is used to estimate anomalies in terms of standard deviations over a period of interest. As for the precipitation and temperature anomaly calculations, 1991–2020 was used as the reference period. Following the recommendations of Stagge et al. (2015), we applied the gamma distribution including a "centre of mass" adjustment for zero precipitation for SPI, and the generalised extreme

value distribution for SPEI. All final SPI and SPEI values were limited to the range -3 to 3 due to the uncertainty related to the extrapolation required for extreme values when based on a limited historical record (Stagge et al., 2015).

### 2.3.2 Wind speed

Hourly eastward and northward components of 10m wind were derived from ERA5-Land hourly data, which is available from 1981 to near-present at a 0.1° regular longitude/latitude global grid (Muñoz Sabater, 2019a) . Daily mean values were
calculated, and the eastward and northward components were combined to derive the daily mean wind speed. Further, the data was remapped using a second order conservative remapping to match the 0.25° grid used for the analysis. For each month, the daily mean wind speed was used to derive the monthly mean wind speed (wpeed_mean), and the 10th and 90th percentile of daily wind speed (wspeed_p10 and wspeed_p90, respectively).

### 2.3.3 Snow and soil moisture

The fractional snow cover and the volumetric soil water in four soil layers were obtained from ERA5-Land monthly averaged data, which is available from 1981 to near-present at a 0.1° regular longitude/latitude global grid (Muñoz Sabater, 2019b). As Fennoscandia covers a wide range of latitudes and altitudes, snow is still present in our dataset for some months and grid cells, although the months analysed were limited to April–October. The volumetric soil water is the volume of water in a given soil layer of the ECMWF Integrated Forecasting System, and is associated with the soil texture, soil depth, and the underlying
groundwater level. The volumetric soil water in soil layer 1 (0–7cm) is one of the best performing datasets of established satellite- and model-based shallow soil moisture products (Beck et al., 2021). As for the wind speed, the fractional snow cover data and soil volumetric soil water data were remapped to a 0.25° grid using a second order conservative remapping. This resulted in four indices: the monthly mean fractional snow cover (snowc) and the monthly mean volumetric soil water in soil layer 1 (0–7cm; swvl1), soil layer 2 (7–28cm; swvl2), soil layer 3 (28–100cm; swvl3) and soil layer 4 (100–289cm;
swvl4). Normalised anomalies of the volumetric soil water indices (swvl1_anomaly, swvl1_anomaly, swvl3_anomaly and swvl4_anomaly) were calculated by subtracting the 1991–2020 mean and dividing by the 1991–2020 standard deviation of each month separately.

### 2.3.4 Land cover

The fraction of burnable area (fraction_burnable) was extracted from the same dataset as used for the satellite-based target
variable (ESA-CCI, 2020). This index represents the fraction of each grid cell that corresponds to vegetated land cover that could burn, i.e. excluding water bodies, permanent snow and ice, urban areas and bare areas. It is based on the Copernicus Climate Change Service (C3S) land cover classes. Details are found in Pettinari and Chuvieco (2018).

## 2.4 The Canadian Fire Weather Index (FWI)

The Canadian Fire Weather Index (FWI; Van Wagner et al., 1987) was used as an alternative fire danger model to compare the performance of the data-driven model with a process-based fire weather index. We chose FWI because it is developed for (Canadian) boreal forests and because it is used for fire danger forecasts in large parts of Fennoscandia (Norway and Sweden: Norwegian Meteorological Institute, 2022; Swedish Meteorological and Hydrological Institute, 2022). Noon temperature, wind speed, humidity and 24-hour precipitation are used to calculate FWI by estimating the moisture content in soil and organic material, fires spread potential and potential heat release in heavier fuel. FWI values are not upper bounded, and the ranges used for classifying fire danger vary. For example, the European Forest Fire Information System (EFFIS) fire danger classes based on daily FWI (San-Miguel-Ayanz et al., 2012) are: very low (<5.2), low (5.2–11.2), moderate (11.2–21.3), high (21.3–38.0) and very high (>=38.0).

The FWI data was obtained from the Copernicus Emergency Management Service (CEMS) global data produced for EFFIS covering 1979 to near-present at a daily resolution (CEMS, 2020). The original 0.25° longitude/latitude grid of the EFFIS dataset is shifted 0.125° as compared to the grid used for the analysis, and a second order conservative remapping was therefore applied to remap the data. Finally, the FWI metrics monthly mean FWI (FWI_mean) and monthly max FWI (FWI_max) were calculated from daily FWI values.

## 2.5 Model set-up and training for Fennoscandia

The target variable and potential predictors comprised the dataset used for the model training. Although a machine learning algorithm for constructing the model is automated, several choices have to be made in a model set-up and training procedure. This section describes these choices and the considerations made, including how to split the dataset into a training and a test set, and the choice of machine learning algorithm, evaluation criterion, complexity terms to be tuned and training procedure specifications.

### 2.5.1 Training and test set

The dataset was split into a training and a test set to obtain an independent evaluation of the final model. Due to temporal dependency in the time series, a fully random selection of data points to the training and test set would likely give a too optimistic evaluation of the model. Instead, assuming limited dependence between years, whole years were selected for both the training and test set. Five years (26%) were selected for the test set: 2004, 2011, 2013, 2017 and 2018. The remaining 14 years (74%) constituted the training set. The years were selected at random, with two exceptions: 1) the year 2018 was manually chosen to be included in the test set to evaluate the model's prediction capability in what is considered an extreme year in terms of hydrometeorological conditions (Bakke et al., 2020), and 2) the year 2017 was randomly chosen for the test set among the years 2016, 2017 or 2019, to allow for comparison with the Norwegian fire occurrence dataset (Sect. 2.6.5) for at least two years (2017 and 2018).

### 2.5.2 Machine learning algorithm

The machine learning algorithm was required to 1) have a straightforward way of estimating the importance of the predictors, 2) have the ability to deal with non-linearities, 3) have the ability to deal with extreme imbalanced data, and 4) have the ability to predict fire danger using probabilities rather than binary classification into fire/no-fire. A random forest classifier (RF) was chosen as a machine learning algorithm that fulfilled the above requirements. The RF, introduced by Breiman (2001), is a model built up of an ensemble of decision tree classifiers (DTs). A DT is a non-parametric supervised learner that builds a tree

by splitting the data multiple times based on predictor values (thresholds) and performs classification estimates based on the target variable values in the end nodes. The complexity of a DT is determined by the tree size and number of predictors. To reduce the variance (instability) of a single DT, an ensemble of DTs can be built based on bootstrap samples of the data and used for prediction by aggregating the DT estimates (bagging). However, the benefit of aggregating the DT estimates is limited by correlation among the DTs. Random Forest is a variant of bagging that meets this shortcoming by randomly selecting a

subset of predictors for each split, hence reducing the correlation among the DTs. The complexity of a RF is determined by the tree size and number of predictors, as well as the number of trees to build. Whereas an increase in tree size and number of (irrelevant) predictors can lead to overfitting, the number of trees to build cannot. However, computational power and computational time limit the number of trees to be built, and the prediction accuracy typically stabilises after a certain number of trees.

Here, we applied the random forest classifier method in the Python package scikit learn (Pedregosa et al., 2011). To control the complexity of the model, the maximum size of each tree (max_depth) and the number of predictors included (Np) were tuned (ref. Sect. 2.5.4). The number of trees was set to 100. The number of predictors to consider for each split was set to the square root of the total number of predictors, as recommended for RF classification problems (Hastie et al., 2009). To account for the imbalance in the target data, the target classes (fire and no-fire) were weighted inversely proportional to

the class frequencies. Remaining parameters were set to default values given by the classifier method. Instead of hardcoded classifications based on the majority class of the end node, probability predictions were calculated based on the proportion of each class. Probability predictions allow for flexibility in the classification threshold, and are needed for the evaluation criterion chosen in this study (Sect. 2.5.3).

### 2.5.3 Evaluation criterion

Model evaluation criteria typically consider all target variable classes with equal weight. Consequently, many traditional evaluation criteria are not applicable due to the extreme imbalance between the number of fire and no-fire data points. As an example for our data, a model predicting no fires at all would have a model accuracy (i.e. proportion of correctly classified data points) of 99.7%, equalling the proportion of no-fires in the target data. This accuracy indicates a near-perfect model for a model that clearly does not meet our objective. Thus, we needed a criterion that is not affected by the extreme imbalance in the target data.

We chose the Area Under the Curve of the Receiver Operating Characteristic (ROC-AUC), which calculates the area under the curve of true-positive rate (Sensitivity) vs false-positive rate (1-Specificity) for different classification thresholds based on the

probability predictions (Fawcett, 2006). Thus, it tackles imbalanced data and takes into account probability prediction in one single measure. A ROC-AUC score less than 0.5 indicates a worse than random model, a value of 1 indicates a perfect model. The ROC-AUC score is 0.5 for a model predicting no fires at all, a model predicting fires for all data points, and (on average for) a model predicting a target dataset in which each data point is randomly selected as fire or no-fire.

### 2.5.4    Training of the model

To avoid overfitting and to make the model as simple as possible without losing prediction capability, the RF complexity parameter controlling the maximum depth of each tree (max_depth) and the number of predictors (Np) were tuned. The max_depth was tuned from values ranging from 1 (the simplest tree structure) to 20 (a complex tree structure). In case of Np, the model was trained to find the best predictor subset using a backward-stepwise selection procedure. To find the optimal combination of max_depth and Np, the model was trained using cross-validation (CV).

The training set was split into seven CV folds by grouping two and two years keeping the number of fires as constant as possible (due to the low number of fires in some years). Full years were selected for the folds to limit the temporal dependency (Sect. 2.5.1). For each CV-iteration, the model was trained on each combination of max_depth and Np. A backward-stepwise selection procedure was implemented to find the best predictor subset for each Np. For each CV iteration and max_depth value, the backward-stepwise selection procedure was as follows: 1) The model was fitted using all predictors, 2) the model evaluation criterion was calculated using the left-out fold, 2) predictor importances were estimated based on the left-out fold, 3) the predictor with the lowest predictor importance was omitted from the predictor subset, and 4) repeating from step 1, now using the new predictor subset instead of all predictors. We also tested an alternative backward-stepwise selection procedure, which explicitly accounted for the high correlation between predictors by omitting the least important of the two most correlated predictors at each step. This method performed less well. A limitation of this method is that predictors with low correlation with other predictors are kept regardless of their importance for the model performance.

Predictor importance was estimated using the permutation importance from the python package scikit learn (Pedregosa et al., 2011), which estimates the decrease in the model score (here: ROC-AUC score) when one of the predictors is randomly shuffled. To increase the robustness of the estimate, the random shuffle was repeated five times, and the mean of the predictor importance of each reshuffling used as the predictor importance estimate. The predictor importance is typically close to zero (or even negative due to randomness) for non-important predictors, as well as for highly correlated predictors. In the case of highly correlated predictors, the predictor importance of one predictor can experience an abrupt increase when a highly correlated predictor is omitted. Thus, the elimination criterion in the backward-stepwise selection cannot depend on a static set of predictor importances (e.g. of the full model; Genuer et al., 2015), but needs to be based on the predictor importances of the updated model for each predictor subset.

### 2.6    Final model selection and evaluation for Fennoscandia

This section describes the selection (Sect. 2.6.1), and evaluation (Sect. 2.6.2–2.6.5) of the final data-driven model for Fennoscandia. The data-driven model performance was evaluated on the test set, and compared with the performance of FWI (Sect. 2.6.2).

Further, the predictor importances were assessed (Sect. 2.6.3), and both the data-driven model and FWI were used to produce fire danger probability maps (Sect. 2.6.4). Finally, the data-driven model was evaluated on an independent dataset, i.e. the Norwegian fire occurrence dataset (Sect. 2.6.5).

### 2.6.1   Selection of a data-driven model

For each combination of max_depth and Np, the average cross-validation ROC-AUC scores were calculated, and the combi-
nation of max_depth and Np yielding the highest score (max_depth_opt and Np_opt, respectively) was selected for the final model. The selection of the optimal predictor subset was not trivial, as the Np_opt predictors selected in each cross-validation iteration could potentially differ due to the varying sub-dataset used as the left-out fold. To select the optimal subset of Np_opt predictors, the seven-fold cross-validation was performed again using max_depth_opt and Np_opt, and the model fitted to each of the predictor subsets selected during the training of the model. The predictor subset yielding the highest average
cross-validation ROC-AUC score was chosen for the final model.

### 2.6.2   Model evaluation

The predictability of the final model was evaluated for the test set (i.e. the years not included in the training of the model) using the ROC-AUC criterion (ref. Sect. 2.5.3). The prediction capability of the data-driven model was then compared with the ROC-AUC of the FWI metrics; FWI_mean and FWI_max (Sect. 2.4).

### 2.6.3   Predictor importance

To estimate the importance of each predictor selected for the final model, we used permutation importance, as described in Sect. 2.5.4, but now using ten random shuffles. For comparison, the impurity-based importances (from the Python package scikit learn; Pedregosa et al., 2011) were also estimated. The impurity-based importance estimates the normalised total re-duction of the criterion introduced by each predictor. Although it is typically biased towards predictors with high cardinality,
typically an issue for numerical predictors, this was not considered a problem here because all input data were continuous. The impurity-based importances sum to 1, and the higher a value, the more important the predictor. As opposed to permutation importance, which can be estimated on any dataset, the impurity-based importance estimates are based on the training set only, which can be misleading in the case of overfitting. On the other hand, in the case of highly correlated predictors, the impurity-based importance may still give a better representation of the importances (as compared to permutation importances)
due to the randomness in the subset of predictors selected for each split in the tree construction. Due to differences in the computation and the pros and cons of the two predictor importance estimates, we applied both the permutation importance and the impurity-based importance. Permutation importances were estimated for the training set and test set separately.

### 2.6.4 Fire danger probability maps

Fire danger probability maps were produced for each month and year in the test set using the prediction probabilities of the data-driven model, and plotted together with fire occurrences from the satellite-based fire occurrence dataset. Similar maps based on FWI_mean and FWI_max were also produced for comparison. In addition, the gridwise Spearman rank correlation between the model predictions, FWI_mean and FWI_max were calculated to reveal any consistent spatial patterns in the agreement (or lack thereof) of the fire danger predictions.

### 2.6.5 Model evaluation using the Norwegian fire occurrence dataset

The Norwegian fire occurrence dataset was included as an independent dataset to evaluate the data-driven model's ability to predict a more detailed fire dataset. The two years included in both the test set for Fennoscandia and the Norwegian fire occurrence dataset (i.e. 2017 and 2018) were used as basis for this evaluation, with Norway as the spatial domain. First, the model ability to predict the original target data (i.e. the satellite-based fire occurrence dataset) and the independent target data (i.e. Norwegian fire occurrence dataset) were compared by computing the ROC-AUC scores. Second, using Norwegian fire occurrence as target, the ROC-AUC score based on the data-driven model was compared to the ROC-AUC scores based on FWI_mean and FWI_max.

Finally, we trained a data-driven model on the Norwegian fire occurrence dataset instead of the satellite-based fire occurrence dataset for Fennoscandia, to get a more "fair" comparison of a data-driven model and the FWI's ability to predict the Norwegian fire occurrence dataset (we note that this step is not included in Fig. 1). Because of the relatively short period covered by the Norwegian dataset, of which 2017 and 2018 were used as test set, only two years (2016 and 2019) constituted the training set. Therefore, a two-fold cross-validation was applied (instead of the seven-fold used in the main analysis) in which each year constituted a fold. The remaining training set-up follows the procedure as described in Sect. 2.5.

### 2.7 Additional experiments

Two additional experiments were performed to test the effect of: (1) using two alternative machine learning algorithms, and (2) not including a dynamical vegetation index as a potential predictor. A description of the experiments are given below.

Two additional, but related, machine learning algorithms were tested to assess the applicability of using the Random Forest algorithm. First, we tested a Decision Tree model (DT; ref. Sect. 2.5.2) to see if a single tree was sufficient to construct a good prediction model. The DT was trained using the same procedure as for the RF model (Sect. 2.5), except that all predictors were evaluated as candidates for each split. Second, we tested a boosting algorithm called AdaBoost (Freund and Schapire, 1997). Boosting algorithms are considered one of the most powerful learning ideas introduced in this century (Hastie et al., 2009). The algorithm makes predictions based on aggregation of results from a selection of constructed weak classifiers (here: DT classifier with max_depth=1). For each new classifier constructed after the first, the data points are weighted based on previous misclassifications, in order to give more emphasis on the misclassified observations. AdaBoost was trained using the same

procedure as for the RF model (Sect 2.5), except that instead of max_depth, the parameter defining the maximum number of weak classifiers to construct (either 50, 100, 200 or 500) was tuned.

We also carried out a separate experiment in which we included a greenness indicator, the Normalized Difference Vegetation Index (NDVI), as a potential predictor. This index is not possible to derive from climate models, and was therefore excluded from the main analysis. However, it is a commonly used index to assess the vegetation status (Smith et al., 2020) and used for fire forecasting (Michael et al., 2021; Chowdhury and Hassan, 2015; Maselli et al., 2003). NDVI can be viewed as a potential estimate of burnable biomass, which is highly variable in the Nordic landscape. NDVI data was obtained from the monthly Terra MODIS Vegetation Indices (MOD13C2) Global 0.05° dataset (Didan, 2015). The data was spatially averaged to a 0.25° resolution. The model training followed the same procedure as described in Sect. 2.5.

## 3 Results

### 3.1 Selection of a data-driven model for Fennoscandia

Based on the average cross-validation (CV) ROC-AUC scores for all combinations of max_depth and number of predictors (Np), the highest score was found for max_depth=9 and Np=15 (Fig. S1). The CV scores revealed that there were no clear best-choice of max_depth independent of the selected number of predictors, and vice versa. This emphasises the importance of testing the combined effect and not fit max_depth and Np separately. For many combinations, the ROC-AUC values were similar (varying only on the second decimal), opening for selecting a simpler model without too high cost of prediction capability. We chose to automatically select the combination yielding the best score, to avoid an extra element of subjectivity. The max_depth of nine was well within the investigated max_depth values. The degrading scores on both ends (i.e. the lowest and highest max_depth values investigated) gave trust in that the range of max_depth values investigated was sufficient. Of the original 30 potential predictors, half remained for the final data-driven model. The number of trees (i.e. 100) used for the data-driven model were considered sufficient, as indicated by the stabilisation of the model performance (Fig. S2).

Of the 15 predictors, five were related to volumetric soil water (swvl1, swvl2, swvl3, swvl4 and swvl2_anomaly), three to temperature (tg_mean, tn_mean and tx_mean), three to wind speed (wspeed_mean, wspeed_p10 and wspeed_p90), two to precipitation (rr_sum and rr_sum_anomaly), and the two predictors snow cover (snowc) and fraction of burnable area (fraction_burnable). All meteorological drought indices and temperature anomalies were omitted. For precipitation and soil moisture, both monthly mean (or sum) values and monthly anomalies were selected. Notably, many of the selected predictors were highly correlated (Fig. 4).

### 3.2 Model evaluation

All ROC-AUC scores presented are computed based on the test set. The final data-driven (Random Forest) model ROC-AUC score was 0.791 (Fig. 5). The data-driven model had a slightly higher ROC-AUC score (differing in the second decimal) compared to the FWI metrics, i.e. monthly max FWI (FWI_max) and monthly mean FWI (FWI_mean). The ROC-AUC scores

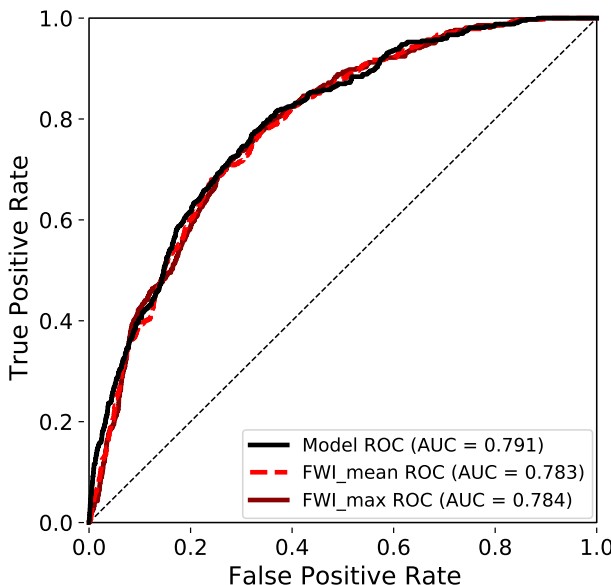

**Figure 5.** Test set ROC curve and ROC-AUC score using the data-driven model as compared to using monthly mean FWI (FWI_mean) and monthly max FWI (FWI_max).

of FWI_max and FWI_mean were more similar, differing in the third decimal (0.784 and 0.783, respectively). The ROC-AUC score of the data-driven model using Random Forest slightly outperformed the two alternative machine learning algorithms tested, Decision Tree (0.737; Fig. S3a) and AdaBoost (0.747; Fig. S4a), confirming that the Random Forest algorithm was a suitable choice for our analysis.

### 3.3 Predictor importance

Figure 6 shows the importances of the predictors, using the permutation importance estimated for the training and test set separately, as well as the impurity-based importance estimates. The dominant predictor in all cases was the volumetric soil water anomaly in soil layer 2 (swvl2_anomaly). The remaining predictors were of significantly less, and similar, importance. In all three algorithms tested, anomaly in shallow volumetric soil water (either in soil layer 1 or 2), monthly mean of daily maximum temperature and volumetric soil water in soil layer 4 were the top three most important predictors (Fig. 6, S3b and S4b). The permutation importance estimates of the training set is generally higher than that of the test set. This is expected since the predictor subset was chosen based on the training set. Differences in the hydrometeorological conditions in the training set years and test set years may also play a role in explaining the differences in the order and magnitude of the permutation importance estimates.

The bivariate and univariate distributions of the most dominant predictor (swvl2_anomaly) and the predicted fire danger probabilities are shown in Fig. 7. The figure shows a clear distinction between fire danger probabilities for fire and no-fire data

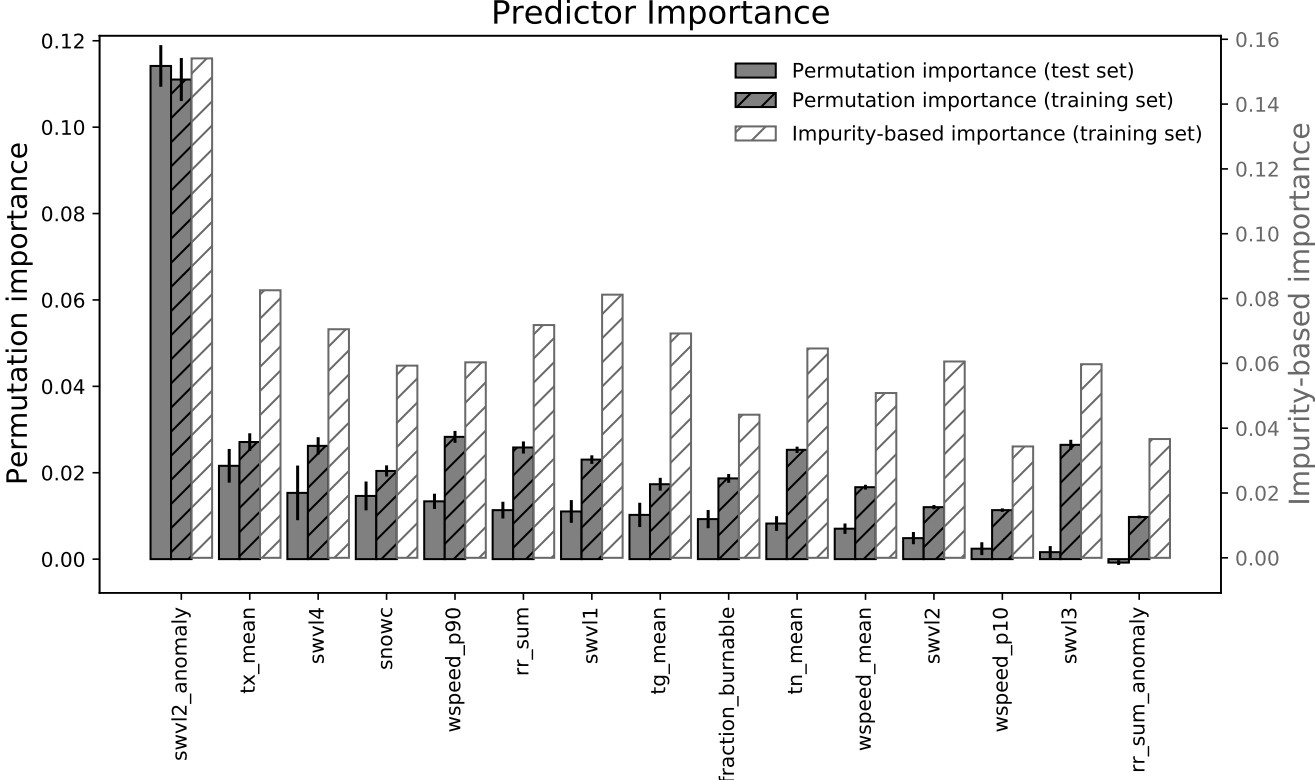

**Figure 6.** The importances of the predictors selected for the final data-driven model estimated using the permutation importance of the test and training set separately, as well as the impurity-based feature importance of the training set. The bar height of the permutation importance represents the mean, and the error-bar represents the standard deviation of ten random shuffles of each predictor. The bars are ordered by the permutation importance of the test set.

points for the training set (Fig. 7a), and less distinct so in the test set (Fig. 7b). For both the training and test set, most fire data points occur for swvl2_anomaly values below zero (dryer soil that normal), and there is a weak negative relationship between the swvl2_anomaly values and fire danger probabilities. However, a high density of fire danger probabilities of approx. zero are distributed along a relatively wide range of swvl2_anomaly values (between approx. -1.5 and 1), pointing to the importance
of other predictors in minimising the fire danger probabilities for these data points.

    In the experiment including NDVI as potential predictor, the test set ROC-AUC score was 0.799 (Fig. S5) compared to 0.791 for the baseline. The most dominant predictor for this experiment was still swvl2_anomaly, with NDVI in second place, regardless of the predictor importance estimate. The final model had a more complex tree structure (max_depth=16) and fewer number of predictors (11) as compared to the model without NDVI. The higher number of predictors in the model without
NDVI can likely be explained by the fact that other predictors are included to compensate for the lack of NDVI information about vegetation health.

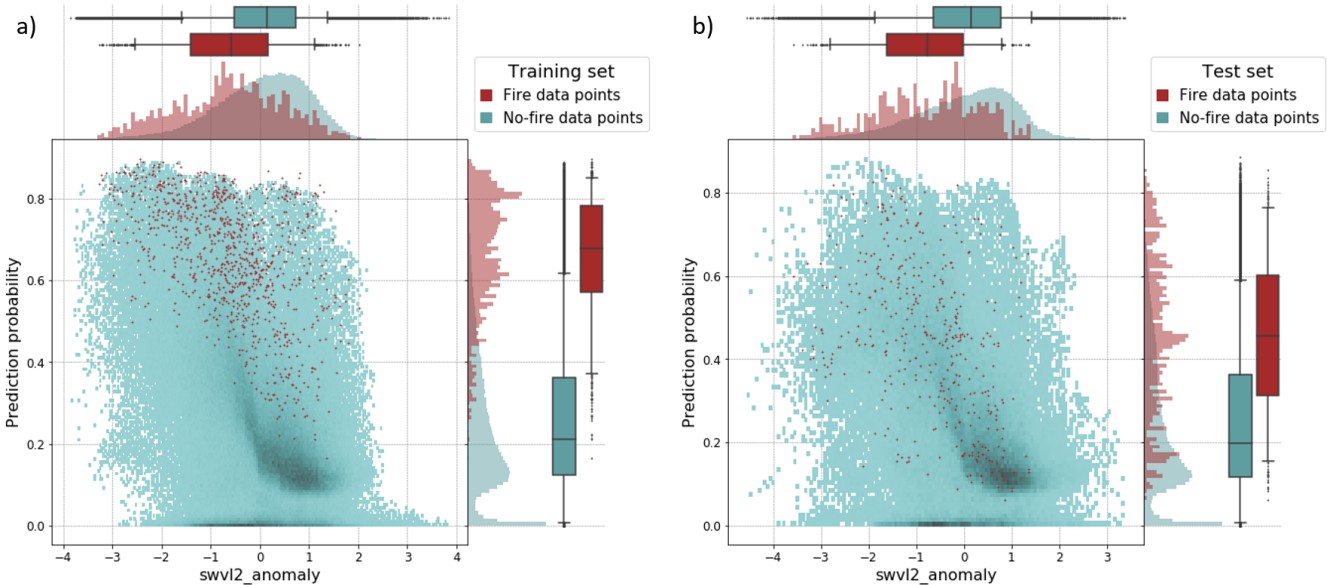

**Figure 7.** The bivariate and univariate distributions of the anomaly in volumetric soil water in soil layer 2 (swvl2_anomaly) and predicted fire danger probabilities. Results are shown for a) the training set, and b) the test set. The distributions are given separately for the fire data points (red) and no-fire data points (blue) from the satellite-based target variable. The histograms show the relative frequencies, and the box plot whiskers define the 5th to 95th percentile range.

## 3.4 Fire danger probability maps

Fire danger probability maps composed using the data-driven model and (for comparison) the FWI metrics for 2018 are shown in Fig. 8, and for the remaining test set years in Fig. S6–S9. The maps also include the actual fire occurrences according to the satellite-based target variable (marked as dots). The year 2018 had one of the highest number of fire occurrences during the period (Fig. 2), which is reflected in high-end fire danger values covering large parts of the region, in particular in May–July as well as August in the southeast. By visual inspection, many of the fires occurred in cells of high fire danger predicted by the data-driven model as well as by the FWI metrics. On the other hand, several months have high fire danger in areas with no fire occurrences. This likely reflects either an actual high fire danger but a lack of ignition sources, or a weakness in the fire danger predictions.

In Fig. 9 high (>0.8) grid-wise rank correlations in fire danger between FWI_max and FWI_mean across Fennoscandia reflect temporal agreement in fire danger between the two FWI metrics. However, the grid-wise rank correlation between the data-driven model and either of the two FWI metrics are spatially less coherent, with approx. 90% of the grid-cell correlations ranging between 0.4 and 0.9. The highest correlations (between 0.7 and 0.9) are found in eastern Fennoscandia (Russia) and along a southwest-northeastern belt from southern Norway, through mid and northern Sweden, to northeastern Finland and Norway. Lowest correlations (<0.5) were found in parts of western Norway, southeastern Sweden and southwestern Finland.

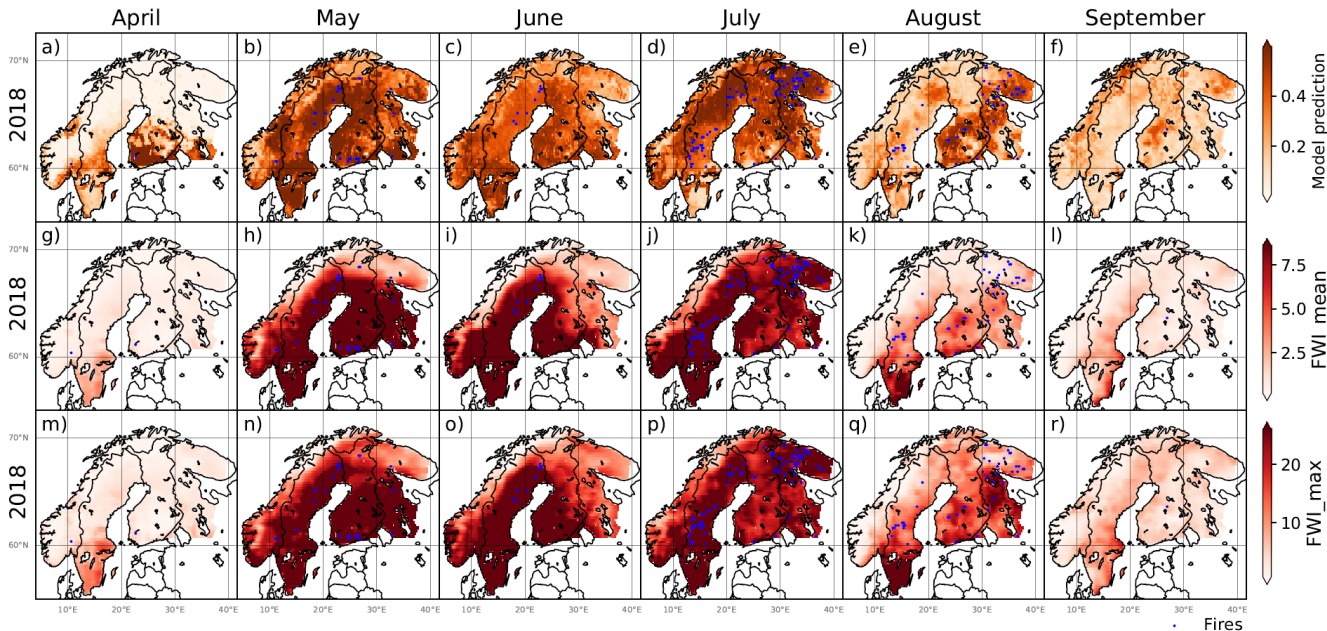

**Figure 8.** Fire danger probability maps for April–September 2018 using a)-f) the data-driven model predictions, g)-l) FWI_mean, and m)-r) FWI_max. Blue markers show fire occurrences using the satellite-based fire occurrence dataset. Colour axes are truncated at the 5th and 95th percentile.

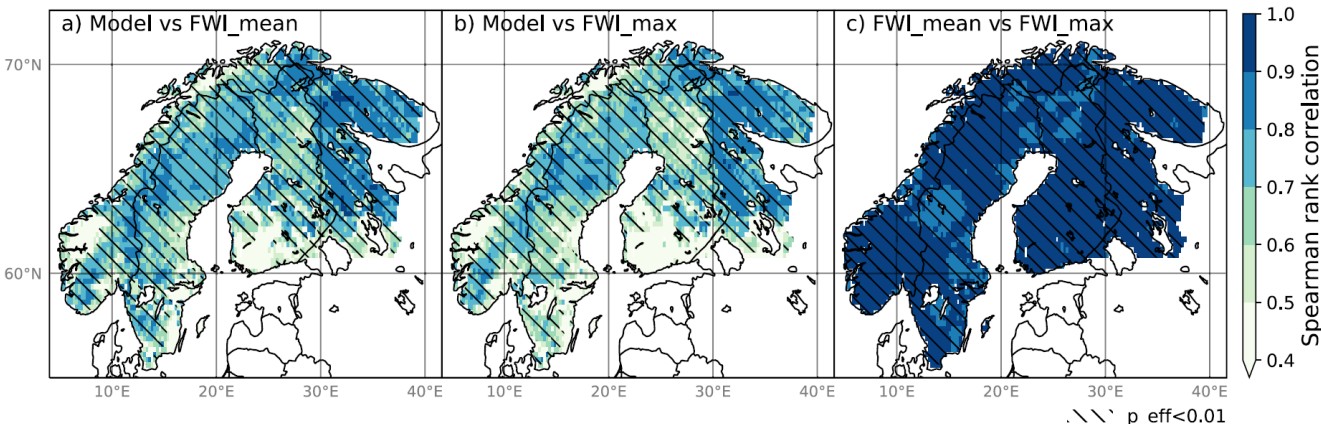

**Figure 9.** Spearman rank correlation of a) the data-driven model and FWI_mean, b) the data-driven model and FWI_max, and c) FWI_mean and FWI_max. The correlations are calculated using the test set. Hatches indicate regions of effective p-value (i.e. p-value accounting for autocorrelation) smaller than 0.01.

## 3.5 Model evaluation using the Norwegian fire occurrence dataset

The ROC-AUC score of the data-driven model prediction of the original (satellite-based) target variable for Fennoscandia was recalculated for the evaluation period (2017 and 2018) with Norway as the spatial domain. The ROC-AUC score of the Norwegian fire occurrence dataset was 0.755 as compared to 0.726 for the satellite-based fire occurrence dataset (Fig. S10a). The pronounced difference in number of fires (Fig. 3) is reflected in the non-smooth ROC-AUC curves of the satellite-based as compared to the Norwegian fire predictions. Due to the low number of fire occurrences in the satellite-based target variable, the corresponding ROC-AUC score must be interpreted with caution.

Prediction of the Norwegian fire occurrence dataset using the two FWI metrics had notably higher ROC-AUC score (0.851 for FWI_max and 0.867 for FWI_mean) as compared to 0.755 for the data-driven model trained for Fennoscandia (Fig. S10b). The monthly fire danger maps for Norway jointly showing fire occurrences, reflect the higher prediction performance obtained by the FWI metrics, in particular in capturing the fire dense areas in May–July 2018 (Fig. S11–S12). The satellite-based target variable had a low representation of fire occurrences in Norway; accordingly, we did not expect an equally good performance of the data-driven model trained for Fennoscandia as for FWI when evaluated for the Norwegian dataset.

The separate data-driven model trained using the Norwegian fire occurrence dataset as the target variable (Sect. 2.6.5) resulted in a final model for Norway with max_depth=2 and Np=8. This represents a considerably simpler model compared to the data-driven model trained using the satellite-based fire occurrence dataset for Fennoscandia. This may be explained by the reduced geospatial complexity in the training set. The ROC-AUC score improved from 0.755 to 0.836 for the test set years 2017 and 2018 (Fig. S13a), although still slightly below the FWI performance. As opposed to the data-driven model trained for Fennoscandia, this model selected monthly maximum temperature and monthly mean of daily maximum temperature as the two dominant predictors (Fig. S13b). The remaining predictors were of notably lower importance, and related to (anomalies in) volumetric soil water, high wind speed, precipitation anomaly and fraction of burnable area.

## 4 Discussion

A data-driven model developed on a monthly and 0.25° spatial resolution was found suitable for fire danger probability mapping in Fennoscandia, despite the region's spatiotemporal heterogeneity in hydroclimatological conditions. In the following, we discuss the selected predictors, known challenges, the added value of a data-driven model and ways forward.

### 4.1 Dominant predictors of Fennoscandian wildfires

The relatively large number of statistically dependent predictors selected for the data-driven model, illustrates the complexity of the controlling mechanisms of fires. In the literature, a wildfire is often referred to as a compound hazard as it is caused by the co-occurrence of several drivers, not necessarily extreme themselves.

The dominant predictor for the data-driven model (both training and test set) was the normalised anomaly of the volumetric soil water in soil layer 2 (swvl2_anomaly). The bivariate plots point to the importance of dryness in the soil relative to normal

conditions for favourable fire conditions. Anomalies in soil moisture are typically a concurrent or delayed response to anomalies in precipitation and evapotranspiration, with the delay depending on subsurface properties such as soil characteristics and depth to groundwater table. This is reflected in a positive correlation between swvl2_anomaly and the meteorological drought indices (SPI and SPEI). Soil moisture has been found a better predictor for burned area than precipitation anomalies in another Boreal region (Baikal region; Forkel et al., 2012), supporting our findings. Studies that report SPI or SPEI as important predictors (e.g. Gudmundsson et al., 2014), often do not include soil moisture anomaly in their study. Our analysis for Fennoscandia finds that soil moisture anomaly is preferred as a predictor over the meteorological drought indices. This is likely due to the direct influence of the soil moisture content on the water uptake by plants and general drying of organic matter, making the biomass more susceptible to combustion. Thus, soil moisture may be considered an indicator of litter fuel moisture conditions. Although swvl2_anomaly stands out as a dominant predictor, the overall weak relationship between this predictor and the fire danger probability as revealed by the bivariate plot, emphasises the importance of other predictors in the data-driven model for Fennoscandia.

Monthly mean daily maximum temperature (tx_mean) and volumetric soil water in the deepest soil layer (swvl4) have the second and third highest test set permutation importance, respectively. Together with anomalies in shallow volumetric soil water (soil layer 1 or 2), these predictors were the most dominant also in the additional experiments using Decision Tree and Adaboost, instead of Random Forest, as the machine learning algorithm. Including NDVI as a potential predictor still gives swvl2_anomaly as the dominant predictor, with NDVI as the second, and swvl4 and tx_mean as the fourth and fifth most important according to test set permutation importance. The consistency in terms of predictors chosen between the experiments emphasises the importance of these predictors in predicting fire danger probabilities. Whereas tx_mean is the average of the highest daily temperatures, affecting the general evaporative demand and transpiration, swvl4 is related to slowly changing deep soil moisture, which is important for water uptake by plants with deep roots. Accordingly, each index has a separate role in controlling the conditions favourable (or non-favourable) for fires, roles that also differ from the role of shallow soil moisture anomaly.

The creation of a data-driven model using the Norwegian fire occurrence dataset as the target variable for the model training, gave a somewhat different selection of dominant predictors. A fewer number of predictors were selected, and the highly correlated tx_max and tx_mean stood out as the two most dominant predictors. We recognise three potential reasons for the difference between this model and the original model developed for Fennoscandia; a change in target variable, study domain and period under investigation, whereof the two latter follow from the first. The target variable was based on ten-fold more fire occurrences for Norway than what was available for the satellite-based dataset for Fennoscandia. It is here worth noting the overall difference in size of the fires recorded in each database, with generally larger fires being represented in satellite-based burned area products than in the Norwegian dataset. Thus, the predictors found for the Norwegian target variable may be more important for small fires, which is typically not included in the satellite-based dataset. Moreover, the difference may reflect different dominant controls of fires in Norway as compared to the remaining part of Fennoscandia, and in particular as compared to the areas with the highest density of fire occurrences in the satellite-based dataset. There is also a likely possibility that the reanalysis data do not represent the volumetric soil water conditions as well in Norway compared to other parts of

Fennoscandia. The difference in period used in the model development should also be considered. In the Norwegian model set-up, only two years were used for training and two for evaluation. These years may not reflect the longer period used for Fennoscandia. In general, more trust is given to models trained on longer time series, enabling a better representation of the variability in hydrometeorological conditions.

Several of the potential predictors derived from ERA5-Land reanalysis (i.e. wind speed, snow cover and soil moisture) were selected as final predictors. This confirms their relevance for predicting wildfires, despite the fact that they combine observations with modelled data. An advantage of reanalysis products over observational datasets is that they are more closely linked to climate model outputs. Thus, the inclusion of reanalysis based indices in the final model for predicting observed fires, points to the prospective of using modelled data for future climate projections. In addition, we found that all ERA5 wind related predictors were selected, while it has been assumed that wind would have a limited impact at these spatial scales (e.g. Aldersley et al., 2011). The wind related predictors may have been selected due to the wind's role in drying of the ground and vegetation by increasing evapotranspiration, its role in spreading the fire to a size recognisable for the satellite, and its indirect role through the link between wind and dominant weather patterns. In short, the selection of ERA5 derived predictors confirms that the use of reanalysis products is useful for wildfire prediction by data-driven model approaches.

### 4.2 Model transferability at the cost of additional potential predictors

Ensuring transferability of the data-driven model to climate projections come at a cost of limiting the type of potential predictors. In the case of predicting fire danger in a stationary climate, several additional predictors are expected to improve the prediction accuracy. Two such predictors are latitude and month of year, which could guide the model to differentiate between important hydrometeorological predictors depending on the season. For example, it is expected that SPEI3 has a different effect on fire danger when the accumulation period covers a snow accumulation period as compared to the growing season. However, such predictors are not suitable for a non-stationary climate as the snow and growing season characteristics are expected to change, e.g. the timing and duration, and the relation to hydrometeorological indices may thus no longer be valid.

Remotely sensed vegetation characteristics have proved useful for predicting burned area on a global scale (Kuhn-Régnier et al., 2021; Forkel et al., 2017). One such predictor is the NDVI, which improved the model to some degree when tested in a separate analysis. Other time varying vegetation or fuel volume indices are expected to further improve the prediction accuracy. However, such data is not always available as continuous spatiotemporal fields, but cover a smaller area over a limited period of time. Climate models that are coupled with DGVMs allow for a wider selection of dynamic vegetation predictors. Vegetation characteristics are found to have a strong relationship with burned area in fire-prone ecosystems (Forkel et al., 2019), and we anticipate that the inclusion of vegetation characteristics available in DGVMs would have improved our model for Fennoscandia.

Other predictors that are expected to improve the fire prediction, are predictors related to sources of ignition. Lightning, sparks from trains, and humans are all important fire starters, and lightning data as well as maps of infrastructure and closeness to human settlement are therefore expected to improve the model predictions. A link between human settlement and fires is not clear from the satellite-based fire occurrence dataset (Fig. 2b). However, the Norwegian fire occurrence dataset (Fig. 3b) sug-

gests a link between wildfire occurrences and population centres. This may partly be due to humans and human infrastructure being fire starters, and partly reflecting an overlap between human settlement in Norway and burnable areas. In addition, the inclusion of ignition sources would have made the model more in line with the target variable (fire occurrences), as the target variable implicitly includes the ignition aspect.

A model constructed for integration with a climate model used for estimating fire probability under different climate scenarios, is different from a model constructed for monitoring/forecasting near-real-time fire occurrences. The two have different application purposes. A model constructed for monitoring/forecasting near-real-time fire occurrences can give a prediction of higher accuracy for short-term preparedness, whereas fire models applicable for use in climate projections are valuable for long-term planning and mitigation strategies.

## 4.3 The effect of the type of fire data chosen as target variable

In this study, we selected a satellite-based fire occurrence dataset as the target variable for the main analysis, and a national fire occurrence record as an alternative target variable for comparison. The lack of small fires in the satellite-based dataset was particularly notable when compared with the Norwegian dataset for April–September 2018 (Fig. 3). During 2018, Norway experienced a record high number of grass and forest fires (DSB, 2019). However, many of these fires were small and rather quickly extinguished, and thus not captured by the satellite. FWI outperformed the data-driven model trained on the satellite-based target variable, in predicting the Norwegian fire occurrence dataset. This was not surprising, as the data-driven model was not trained on the Norwegian dataset, small fires in general, and for most of Norway, no fires at all. A data-driven model trained on small fires in Norway considerably improved the prediction ability, despite that only two years were available for model training.

It is not given which of the data sources for fire occurrence is better to use as target variable. A benefit of national records is that they typically have registered most fire occurrences, including small fires. However, the recording procedures and information logged may have changed over time, and vary from country to country (e.g. Aalto and Venäläinen, 2021). In addition, there are large differences among countries in the coverage of historical fire recording, and the availability of such datasets, limiting transnational studies. Satellite-based burned area products are typically readily available, consistent across country boundaries and exist for longer periods than what can be found for many national recordings. They also have a global spatial coverage, which allows for a large-scale application of the proposed methodology, such as an analysis of different drivers in different regions.

As an alternative to burned area, satellite-based active fire products can be used to construct a fire occurrence dataset. Active fire products are capable of detecting smaller fires compared to standard burned area products (Wooster et al., 2021; Oliva and Schroeder, 2015). Whereas small fire detection is improved in many regions by using active fire products, detection errors (i.e. false fires) are a problem in some regions and seasons (Wooster et al., 2021; Zhang et al., 2018). The active fire products detect burning at the time of overpass given relatively cloud-free conditions, which can be a problem for regions within Fennoscandia that are seldom cloud-free. We chose to apply the burned area product because it is considered less sensitive to cloud-cover.

Further, the burned area product have a more direct relevance to climate-relevant consequences, such as albedo and ecosystem functioning. In addition, an independent target dataset was included for comparison, i.e. a local fire record of Norway.

Whether or not the lack of small fires in the satellite-based products is a limitation or not, depends on the objective. For example for forecasting, monitoring or projections used for fire preparedness planning in Norway, capturing small fires is vital as small fires have the potential to develop into large fires with devastating impacts. It is worth noting that small fires are not

necessarily small following natural conditions, but may be so following the wildfire preparedness and suppression in the area. Thus, predicting small and large fires may be of similar importance for a region. However, fires that stay small, are of less importance in terms of large-scale changes in emissions, albedo and ecosystem functioning.

### 4.4 Fire danger probability mapping

To our knowledge, our study is the first in which a data-driven model is developed for Fennoscandian wildfire danger, by means

of training on transnational datasets derived from satellite imagery over multiple years at a sub-yearly time step. The spatial and temporal resolution of the data-driven model presented in this paper takes into account the variable hydrometeorology over the region, seasons, and years, which is necessary in order to make use of the model to produce fire danger probability maps.

The present study confirms that both the FWI and our data-driven model are skilful models for fire danger probability mapping in Fennoscandia. The ROC-AUC scores were relatively high for both; especially given the lack of ignition in both

models. The good performance was reflected in the general ability of the fire danger probability maps to predict high fire danger in regions were fires occurred. High fire danger probabilities are also found in data points without fire occurrence. This was expected, as ignition is needed for a fire to occur. The varying grid-wise rank correlation between the data-driven model and each of the two FWI metrics (Fig. 9) underscores that fire danger probability maps produced by the two different approaches are different despite their similar and skilful overall performance. An interesting spatial pattern, is the notable difference in

correlation closely following the Russian-Finnish border, with the higher correlations found in Russia. A likely reason for this is the fact that the data-driven model is better tuned to Russian conditions as compared to Finnish conditions due to the relatively higher number of fires in Russia (Fig. 2), whereas the FWI performance is independent of the fire occurrence density. The spatially varying correlation between FWI and the data-driven model, highlights the benefit of including different types of models to improve our knowledge of the uncertainties related to fire danger. Thus, we do not suggest replacing current process-

based models with data-driven models, but recommend using them jointly in assessments of fire occurrences (or probability thereof).

### 4.5 Added values of a data-driven model and ways forward

A data-driven (statistical/machine learning) model differs fundamentally from a process-based global fire model or fire weather index. Process-based models use established or assumed relationships between various indices and fire occurrence to construct

fire danger or fire occurrence models. While a data-driven model is also based on process understanding in the selection of indices tested as potential predictors, it differs by explicitly accounting for fire occurrences in the construction the model.

There are several benefits of a data-driven approach for mapping fire danger probability. First, a data-driven model can be applied as an additional and independent model, as already mentioned in Sect. 4.4. It can either be fire danger probability mapping as exemplified in this study, or it can be combined with ignition and/or spread, for example in a similar way as is done in process-based global fire models. Depending on the construction of the model, it can be applied jointly with both fire weather indices and process-based global fire models. Assessments using multiple models of fundamentally different construction can improve the trust in the predictions, and the knowledge of the prediction uncertainties.

Second, a data-driven approach can automatically sort out important predictors, and omit the remaining. In this way, they can provide useful new insight into which indices one should consider when analysing the probability of fire occurrence. For example, soil moisture data is usually not considered in fire weather indices such as FWI, whereas shallow soil moisture anomaly was found the most dominant predictor by the data-driven model for Fennoscandia. New insight into relevant predictors can help improve the process-based models. One can also investigate the important predictors' sensitivity to spatial and temporal resolutions as long as the potential predictors and target variable allow it. A finer spatiotemporal resolution may reveal other (fine-scale) indices, such as altitude and local wind, as dominant predictors, as compared to a coarser resolution where such indices are averaged out.

A benefit of a data-driven model as developed here, is that the approach can be transferred to other regions. This is in opposition to FWI, which is developed for boreal forests, and should be used with caution when applied for other biomes (Bedia et al., 2018; Dowdy et al., 2009). In this work we used a satellite derived fire dataset and globally available hydrometeorological variables, which can all be obtained for other regions around the world. This would only require additional training on the local settings, but would not require a new workflow or model implementation. This regional transferability can be combined with the flexibility of implementing different sets of potential predictors and target variables, allowing for regional and application specific analysis. This was exemplified by the present study, in which a data-driven approach was applied for predicting fire danger probabilities on a understudied region of the boreal region with highly varying hydroclimatology. A notable improvement was found when using the local fire occurrence dataset to train the model, illustrating the potential of high-performing data-driven models adapted to local conditions, when high-quality target data is available. The difference in the selection of dominant predictors between the data-driven model constructed for Fennoscandia and Norway, exemplified that certain relationships are more important in some regions than others. The flexibility in spatial domain and predictors also allows for large-scale analysis, more in line with process-based global fire models in terms of constructing regional-independent relationships between drivers and fires.

By limiting the potential predictors to those available in most climate model when developing the data-driven model, the final model allows for analyses of future changes in fire danger probabilities given different climate scenarios. Although outside of the scope of this study, applying our model in future climate scenarios can give valuable new insight as what to expect of changes in fire danger probability in Fennoscandia under future climate scenarios.

In summary, our study has demonstrated the value of a data-driven model as an independent model for constructing monthly fire danger probability maps, and as a tool for identifying dominant predictors. Data-driven models have a high degree of flex-

ibility making them suitable for adaptation to other regions and applications. Thus, we regard data-driven models as valuable contributions in a wide range of applications related to fire monitoring, forecasting and projections.

## 5    Conclusions

The data-driven approach was found suitable to identify dominant predictors for fire occurrence and to construct spatiotemporal
resolved fire danger probability maps in Fennoscandia. Anomalies in the volumetric soil water in soil layer 2 (7–28 cm) were found to be the dominant predictor, followed by monthly mean of daily maximum temperature and volumetric soil water in the deepest soil layer (layer 4; 100–289 cm). Other selected predictors were related to wind speed, precipitation, snow cover and fraction of burnable area. The selected predictors emphasise the importance of other predictors than weather alone, as has traditionally been used for fire weather indices. In addition, the variation in the type of predictors emphasises the complexity
in driving mechanisms for fire occurrence and the value of a bottom-up approach to automatically identify the most important predictors.

The following concludes our research questions presented in the introduction:

1. The data-driven model for Fennoscandia was comparable to (and slightly outperforming) the Canadian Fire Weather Index (FWI), which was developed for similar biomes and latitudes as Fennoscandia. The temporal rank correlations of
the fire danger probability maps produced by the two approaches showed large spatial variability, pointing to the value of including more than one approach when mapping fire danger.

2. The data-driven model performance decreased, and was outperformed by the FWI, when used to predict a local fire occurrence dataset for Norway. This can be explained by the lack of fire occurrences in Norway in the satellite-based target variable used for model training.

3. When using the Norwegian fire occurrence dataset as target variable in the training, the model performance increased and reached a similar performance as for FWI, despite that only two years were available for the model training.

4. The Random Forest algorithm used in the main analysis outperformed the simpler (Decision Tree) and more sophisticated (AdaBoost) machine learning algorithm. The Random Forest algorithm was therefore found suitable for the objective of this study. Nevertheless, we acknowledge the potential of yet other machine learning algorithms not tested here, to
improve the predictions further.

5. There was a minor decrease in the model performance when NDVI was not included as a potential predictor. Thus, most of the effect of NDVI for fire occurrence is compensated for by other predictors. In a monitoring or forecasting situation, where the transferability to climate models is not important, the inclusion of NDVI can be useful.

The selection of potential predictors was limited to predictors available in most climate models and transferable to different
climate scenarios. Accordingly, our model allows for analyses of future changes in fire occurrence characteristics, which would be a natural next step. This can preferably be done jointly with process-based approaches, in order to evaluate the agreement

and spread among the different types of models. The approach presented in this study can also be adapted to other regions and with the inclusion of other potential predictors.

Finally, we want to make a general remark on the importance of user-made choices in data-driven approaches. Even though the machine learning (or statistical) algorithm in itself is automated, the model construction and estimated performance can be highly sensitive to user-made choices. Examples include choices of the machine learning algorithm, training procedure, selection of target variable and potential predictors, predictor subset selection, evaluation criterion (in particular in the case of extreme imbalance), and the importance of testing the model on a dataset independent of the model construction. In the present study, we aimed to make all user-made choices transparent and justified, and we tested for alternative options for several of the choices. In conclusion, a data-driven approach has proven an important tool to identify dominant predictors of fire occurrence and as an alternative fire danger probability model to already established process-based models, as this study demonstrates.

*Code availability.* Code is available upon reasonable request to the corresponding author. Command line *Climate Data Operators* (CDO; Schulzweida, 2021) and the Python package *xarray* (Hoyer and Hamman, 2017) were used for processing the NetCDF files. SPI and SPEI calculations were performed using the *SCI* package in R (Gudmundsson and Stagge, 2016). The remaining calculations and visualisations were performed using Python: *NumPy* (Harris et al., 2020) and *pandas* (Pandas development team, 2020) were used for data handling, *Scikit-learn* (Pedregosa et al., 2011) for the construction, training and evaluation of the data-driven models, *SciPy* (Virtanen et al., 2020) for computation of the correlations between predictors, *xskillscore* (xskillscore, 2021) for calculating the correlation effective p-value between the data-driven models and FWI metrics, and *Matplotlib* (Hunter, 2007), *cartopy* (Met Office, 2010 - 2015) and *seaborn* (Waskom, 2021) for the visualisations of the results.

*Data availability.* All data used are available online (urls accessed 16.06.22): E-OBS (https://doi.org/10.24381/cds.151d3ec6), ERA5-Land (https://doi.org/10.24381/cds.151d3ec6 and https://doi.org/10.24381/cds.68d2bb30) and v5.1.1cds (https://doi.org/10.24381/cds.f333cf85) are available at the Copernicus Climate Change Service (C3S) Climate Data Store, NDVI data at NASA's Land Processes Distributed Active Archive Center (LP DAAC; e4ftl01.cr.usgs.gov/MOLT/MOD13C2.006/), and the Norwegian wildfire record at the services by Norwegian Directorate for Civil Protection (DSB; http://www.brannstatistikk.no/). Note that the DSB webpage is in Norwegian. Data are freely available, and in case of any questions regarding the data, please use the contact information provided by the webpage.

*Author contributions.* SJB and LMT designed the study. SJB performed the analyses and visualisations, and drafted the paper. All authors critically discussed the results, and reviewed and edited the draft.

*Competing interests.* The authors declare that they have no conflict of interest.

*Acknowledgements.* We greatly acknowledge all data providers. ERA5-Land and v5.1.1cds data were downloaded from the Copernicus Climate Change Service (C3S) Climate Data Store, and NDVI data was downloaded from NASA's Land Processes Distributed Active Archive Center (LP DAAC). We acknowledge the Norwegian Directorate for Civil Protection (DSB) for the Norwegian wildfire record. We acknowledge the E-OBS dataset from the EU-FP6 project UERRA (https://www.uerra.eu) and the Copernicus Climate Change Service, and the data providers in the ECA&D project (https://www.ecad.eu). We thank Monica Ionita and Trond Simensen for valuable input. Niko Wanders acknowledges funding from NWO 016.Veni.181.049.

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

**Table 1.** Target variable (fire occurrence), and potential predictors included in the main analysis. All normalised anomalies are calculated based on each corresponding month's mean and standard deviation from the reference period 1991–2020. The same reference period was used when calculating SPI and SPEI.

| Category | Name | Unit | Description | Source dataset |
|---|---|---|---|---|
| Fire occurrence | Satellite-based fire occurrence | Class | Target variable of the main analysis. Classification of burned area: class 1 (fire) if burned area>0, otherwise 0 (no-fire) | v5.1.1cds 0.25° lon/lat |
| | Norwegian fire occurrence | Class | Target variable for comparison. Classification of fire record: class 1 (fire) if fire occurred, otherwise 0 (no-fire) | Norwegian record point-based |
| Precipitation | rr_sum | [mm] | Monthly precipitation sum | |
| | rr_sum_anomaly | [-] | Anomalies of rr_sum | |
| Temperature | tg_mean, tn_mean and tx_mean | [°C] | Monthly mean of daily mean, daily minimum and daily maximum temperature | EOBS v23.1e 0.25° lon/lat |
| | tx_max | [°C] | Monthly maximum of daily maximum temperature | |
| | tg_mean_anomaly, tn_mean_anomaly and tx_mean_anomaly | [-] | Anomalies of tg_mean, tn_mean and tx_mean | |
| Meteorological drought | SPI2, SPI3, SPI6 and SPI9 | [-] | SPI {-3,3} over 2, 3, 6 and 9 months, calculated from rr_sum | |
| | SPEI2, SPEI3, SPEI6 and SPEI9 | [-] | SPEI {-3,3} over 2, 3, 6 and 9 months, calculated from rr_sum minus monthly potential evapotranspiration, calculated based on tg, tn and tx | |
| Wind speed | wspeed_mean | [m/s] | Monthly mean 10m wind speed. | ERA5-Land hourly 0.1° lon/lat |
| | wspeed_p10 and wspeed_p90 | [m/s] | Monthly 10th and 90th percentile of daily 10m wind speed. | |
| Snow | snowc | [-] | Monthly average fraction of grid cell occupied by snow. | ERA5-Land monthly 0.1° lon/lat |
| Soil moisture | swvl1, swvl2, swvl3 and swvl4 | [m$^3$/m$^3$] | Monthly mean volumetric soil water in soil layer 1 (0–7 cm), layer 2 (7–28 cm), layer 3 (28–100 cm) and layer 4 (100–289 cm) | |
| | swvl1_anomaly, swvl2_anomaly, swvl3_anomaly and swvl4_anomaly | [-] | Anomalies of swvl1, swvl2, swvl3 and swvl4 | |
| Land cover | fraction_ burnable | [-] | Fraction of the cell corresponding to vegetated land covers that could burn. | v5.1.1cds 0.25° lon/lat |