# Peer review of "A data-driven model for Fennoscandian wildfire danger"

_Natural Hazards and Earth System Sciences, 2021_

## Author Response (AR1)

Dear Editor,

Thank you for taking your time to handle and report on our manuscript. Hereby, we would like to provide our point-by-point reply to the comments of Referee #1 (RC#1) and Referee #2 (RC#2). Finally, we also respond to the remarks from the editor.

Original comments are marked by the referee abbreviation 'RC#1' or 'RC#2', the remark by the editor by 'Editor', our responses by 'Authors' and reference to the places where changes have been made in the track-changes version of the manuscript are marked by 'Change'. We have also added changes in grey text under "Change" to make it easy for the reader to see the changes made in relation to our answer. If no changes were made following the comment, we write "No change in manuscript." under 'Change'. In addition to the changes following the comments by RC#1, RC#2 and Editor, we have made minor text editions to correct spelling/grammar or increase readability. All text changes are visible in the track-changes version of the manuscript. Note that the track-changes file does not mark changes for remade figures. All figures of maps have been changes to include latitude and longitude (Fig. 2, 3, 8, 9, S6, S7, S8, S9, S11 and S12).

Kind regards,

Niko Wanders, Karin van der Wiel, Lena M. Tallaksen and Sigrid J. Bakke

**Response to comments by Referee #1 (RC#1)**

| 1.01 | RC#1 | This paper uses Random Forests to estimate wildfire probability in the mostly boreal Fennoscandia region. Comparable studies using similar data and Random Forest models have been performed over various spatial domains but this study is the first one focusing on Fennoscandia in particular. The analyses are thorough and very well documented. There are a few issues I would like to see addressed before publication: |
| --- | --- | --- |
| | | What was the motivation to perform the analysis at a 0.25° and not the native MODIS resolution, or at least at the finest meteorological resolution? You lose a lot of spatial detail in this way. Pixel product data are available at a 250 m resolution. |
| | Authors | We chose the 0.25 degree resolution to investigate if a data-driven model is applicable for use in combination with the current state of the art global climate models, rather than aiming for the highest spatial resolution possible. Further, spatial dependency of fires (e.g. the same fire occurring in two or more cells) is reduced when using a coarser scale. We see that the reasoning behind the spatial scale chosen is not stated clearly in our manuscript, and we have clarified it in the revised version. |
| | Change | Line 168-171 |

"The fire burned area dataset is available both as a 0.25° longitude/latitude regular grid product and as a pixel product of 250 m resolution. We chose to use the grid product to investigate if a data-driven model is applicable for use at the spatial scale of the state of the art global climate models. Further, spatial dependency of fires (e.g. the same fire occurring in two or more cells) is reduced when using the coarser scale of the grid product as compared to the pixel product."

| 1.02 | RC#1 | Not including dynamic vegetation predictors or specific land cover is a weakness. Recent work (e.g. Kuhn Regnier et al., 2021) has shown that adding vegetation dynamics has considerable impact on model skill. NDVI not being modelled by DGVMs is not a valid justification as several productivity-related indicators estimated by DGVMs are available from Earth observation. The same applies to (more static) land cover information, such as crop fraction or tree type (e.g. Forkel et al., 2019). |
|---|---|---|
| | Authors | Thank you for the references. As you state, several productivity-related indicators are estimated by DGVMs. Still, most climate model outputs are not based on runs for which the climate model is coupled with a DGVM. For this reason, we wanted to limit the choice of predictors to those available from climate models without the need of DGVMs. We have clarified our reasoning, and acknowledged the possibility of productivity-based indicators estimated by DGVMs in the revised manuscript. |
| | Change | Line 213-217 and line 604-607 |

"Dynamic vegetation related predictors were excluded because most climate model outputs are not based on runs for which the climate model is coupled with a Dynamic Global Vegetation Model (DGVM), but rather use prescribed vegetation cycles." and rearranged the preceding sentence.

"Climate models that are coupled with DGVMs allow for a wider selection of dynamic vegetation predictors. Vegetation characteristics are found to have a strong relationship with burned area in fire-prone ecosystems (Forkel et al., 2019), and we anticipate that the inclusion of vegetation characteristics available in DGVMs would have improved our model for Fennoscandia."

| 1.03 | RC#1 | The same applies to socio-economic drivers such as population density. Fig.3 suggests that there is clear link between wildfire occurrence and population centres. Probably, including crop fraction as variable would already be a good proxy for this. |
|---|---|---|
| | Authors | We agree that socio-economic predictors would likely improve the model prediction. The main reason that we hypothesise this is that humans and human infrastructure are fire starters (line 608-610). Figure 3b of Norway suggests a link between wildfire occurrence and population centres. We suggest this is partly due to humans being a major ignition source as already mentioned, as well as the geographical overlap between human settlement in |

Norway and burnable areas (a potential predictor included). However, as seen in Figure 2, which shows number of fires over Fennoscandia, the link between human settlement and fires is not clear. We chose to constrain our study to predictors available in global climate models. In a future study it would have been of interest to test the inclusion of socio-economic and vegetation based predictors, however, this is beyond the scope of the current study (also as one aim is to compare with the FWI, which neither accounts for socio-economic predictors). We have made a comment in accordance with our answer in the revised manuscript.

Change      Line 610-614

"A link between human settlement and fires is not clear from the satellite-based fire occurrence dataset (Fig. 2b). However, the Norwegian fire occurrence dataset (Fig. 3b) suggests a link between wildfire occurrences and population centres. This may partly be due to humans and human infrastructure being fire starters, and partly reflecting an overlap between human settlement in Norway and burnable areas. In addition, the inclusion of ignition sources would have made…"

**Detailed comments:**

1.04   RC#1      The title is a bit misleading: The model identifies the main hydrometeorological drivers of past wildfire occurrences. It estimates the probability of wildfire occurrence but it does not predict (i.e. forecast) wildfire occurrence itself; It this should be made clear in the title.

       Authors      We agree, and revised the title to "A data-driven model for Fennoscandian wildfire danger". We have removed "prediction" from the text where we see that it can be misunderstood as a synonym for forecasting. We still use the term 'prediction' in the context of machine learning, which refers to the output of a data-driven model (here fire danger probability).

       Change      Changed the title. Removed 'prediction' from the text in line 14, 16, 121, 150, 207, 281, 483, 486, 504, 505, 520, 534, 552, 553, 616, 617, 618, 620, 654, 671, 679, 680, 712 and 758.

       Note      New title: "A data-driven model for Fennoscandian wildfire danger"

1.05   RC#1      l68-69: It is misleading to state that fire-weather indices based on climate model and reanalysis data can be used for monitoring and forecasting.

       Authors      We agree and have rephrased these sentences.

       Change      Line 72-77

"Fire weather indices can also be calculated based on large-scale gridded reanalysis and climate model data (e.g. McElhinny et al., 2020), allowing for spatially continuous estimates. Such estimates are used for assessments of historical and future changes in fire danger (Sun et al., 2019; Abatzoglou et al.,

2019; Jolly et al., 2015; Flannigan et al., 2013), whereas fire weather indices calculated using numerical weather forecast models are used for transnational fire monitoring and forecasting (San-Miguel-Ayanz et al., 2012)."

| 1.06 | RC#1 | l91: mention some of these limited studies using data-driven methods to predict intra- and inter-annual dynamics, e.g. Forkel et al. (2017, 2019) and Kuhn-Regnier et al. (2021), who predict monthly global patterns. |
| | Authors | Thank you for providing these references. We have included these in the revised manuscript. |
| | Change | line 98-101 (and mentioned the references in line 55, 601 and 606) |

"Data-driven model studies accounting for both seasonal and inter-annual variability in fire occurrences are limited. Those that exist typically predict monthly global patterns in burned area using predictors from observational data (Forkel et al., 2017), DGVMs (Forkel et al., 2019) or a combination of observational and reanalysis data (Kuhn-Régnier et al., 2021)."

| 1.07 | RC#1 | l92: How do you define a data-rich region? With recent satellite availability, practically all regions have become data rich and several studies ha |
| | Authors | We agree this is an unclear statement and have clarified it in the revised manuscript. The last part of your comment is unfortunately lacking, however, we trust your key point is clear from this sentence part. |
| | Change | Line 102-103 |

"Data-driven methods are restricted to regions and applications that have sufficient data to both train the models and validate their performance."

| 1.08 | RC#1 | l97: "In addition, a bottom-up approach is typically less straightforward in its data requirements and methodology as compared to the process-based approaches" -> explain |
| | Authors | We have clarified this in the revised manuscript. |
| | Change | Line 109-110 |

"…, because a bottom-up approach is not limited by the physical understanding of the system, and the amount of data and algorithms implemented are in principle unlimited."

| 1.09 | RC#1 | l123: unclear whether this dataset is used for training or as independent validation reference. If used as target in model development, this doesn't come out clearly in Fig.1 (as it should also be split up into training and testing) |

| Authors | The local (Norwegian) fire occurrence dataset is used as an independent validation reference for research question 2, and used for training (a target in model development) for research question 3. For research question 3, we split up the Norwegian fire occurrence dataset into training and test datasets. This is described in line 421-422. Figure 1 shows the data-driven approach for the Fennoscandian domain (i.e. the one developed using satellite-based fire data as target), and not that of the Norway alone. It is stated in the figure caption. Further, we have made a line break after punctuation in line 375, to separate the two applications of the Norwegian dataset. We considered including this additional analysis in Figure 1, but concluded it would make the figure more messy than clarifying, however a note is made in the text (stated that this analysis is not included in the figure). |
|---|---|
| Change | Line 148, line break after line 418 (not marked as change), and parenthesis in line 421. |
| | "A general outline of the data-driven approach for Fennoscandia is shown in Fig. 1." |
| | "… (we note that this step is not included in Fig. 1)." |

| 1.10 | RC#1 | l127: How can the machine learning algorithm both be simpler and more sophisticated? |
|---|---|---|
| | Authors | The simpler and more sophisticated machine learning algorithms are two separate algorithms (Decision tree is the simpler and AdaBoost is the more sophisticated one). We have clarified it by rephrasing the sentence. |
| | Change | Line 141-142 |
| | | "4. Does the data-driven model chosen outperform both a simpler machine learning algorithm (Decision Tree), as well as a more sophisticated (AdaBoost) machine learning algorithm?" |

| 1.11 | RC#1 | l181: Why are dynamic vegetation predictors not included? Recent work (e.g. Kuhn Regnier et al., 2021) has shown that adding vegetation dynamics ha considerable impact on model skill. |
|---|---|---|
| | Authors | With reference to our earlier comment on the subject (in our answer to your question 1.02 regarding DGVM), we have clarified our reasoning in Sect. 2.2 in the revised manuscript. |
| | Change | See comment 1.02. |

| 1.12 | RC#1 | l220: why is wind speed included as predictor? More a predictor of fire spread than of occurrence |
|---|---|---|
| | Authors | For a fire to occur in the burned area dataset, it must have been of a size recognisable for the satellite. Thus, the fire must have spread to some degree |

(due to wind or not). Another effect of the wind is drying of the ground and vegetation by increasing evapotranspiration prior to the fire. Regardless of the reason, wind was found to be a selected predictor, indicating its importance in predicting the fire occurrence dataset. We have added a comment about this in the revised manuscript.

| Change | Line 587-590 |
|---|---|

"The wind related predictors may have been selected due to the wind's role in drying of the ground and vegetation by increasing evapotranspiration, its role in spreading the fire to a size recognisable for the satellite, and its indirect role through the link between wind and dominant weather patterns. In short, the selection of ERA5 derived predictors confirms…"

| 1.13 | RC#1 | l314: Which threshold was used beyond which no more predictors were removed? |
|---|---|---|
| | Authors | We used no threshold; a predictor subset was made for all (each) number of predictors (Np), as described in Sect. 2.5.4. This can also be seen in Fig. S1, which shows the average cross-validation score for each combination of max depth and number of predictors from one to all (30) predictors. The Np selected for the final model was selected as described in Sect. 2.6.1. |
| | Change | No change in manuscript. |

| 1.14 | RC#1 | l394: Why did you not assess the impact of a predictor that is |
|---|---|---|
| | Authors | Unfortunately, the last part of your comment is missing. |
| | Change | No change in manuscript. |

| 1.15 | RC#1 | Section 3.1/l415: The final set of predictors, which mostly excludes anomaly-based indicators, seems to suggest that the model is tuned to predict fire occurrence climatology rather than typical fire weather situations. Is this correct? |
|---|---|---|
| | Authors | We do not fully agree that the model predict fire occurrence climatology rather than typical fire weather situations. First; even though most of the anomaly-based potential predictors are not included in the final set of predictors, the shallow soil water anomaly stands out as a clear dominant predictor as compared to the other selected predictors. Secondly, the predictors have a high annual variability in monthly values. Notable differences from year to year for the same month can be seen in the fire danger probability maps produced by the model (in Fig. 8 and S6-S9), for example July 2017 (Fig. S9d) versus July 2018 (Fig. 8d). |
| | Change | No change in manuscript. |

| 1.16 | RC#1 | L419-421: is the minor difference between the RF model and the FWI predictors really significant? |
|---|---|---|
| | Authors | We did not test for significance, but we agree that this difference is likely not significant. We have changed to a more precise language (e.g. at which digit they differ) in the revised manuscript. |
| | Change | Line 465-469 |

"The data-driven model had a slightly higher ROC-AUC score (differing in the second decimal) compared to the FWI metrics, i.e. monthly max FWI (FWI_max) and monthly mean FWI (FWI_mean). The ROC-AUC scores of FWI_max and FWI_mean were more similar, differing in the third decimal (0.784 and 0.783, respectively). The ROC-AUC score of the data-driven model using Random Forest slightly outperformed the two…"

| 1.17 | RC#1 | l442-447: To me it's not very surprising that simply including NDVI does not improve model skill as it's climatology closely follows that of soil moisture and meteorological variables. Did you also test the inclusion of NDVI anomalies? |
|---|---|---|
| | Authors | We did not include NDVI anomaly. NDVI can be viewed as a potential estimate of burnable biomass (in particular in the Nordic landscape that has a high variability in burnable biomass) and it is therefore preferred to include the absolute NDVI value instead of the NDVI anomaly. The close relationship between NDVI and hydrometeorological variables, such as temperature and snow cover, further argues for developing models without NDVI. As acknowledged earlier, many more variables could have been included in our study (NDVI anomaly being one of them), however, some constrains in the number of predictors included had to be made at the start of our study. We have added a comment about that NDVI can be viewed as a potential estimate of burnable biomass, which is highly variable across space and time in the Nordic landscape. |
| | Change | Line 441-442 |

"NDVI can be viewed as a potential estimate of burnable biomass, which is highly variable in the Nordic landscape."

| 1.18 | RC#1 | l445: High fire danger (luckily) most of the times does not lead to actual wildfire activity as an ignition source is required. |
|---|---|---|
| | Authors | We agree. The relation to line 445 (i.e. line 492 in the track-changes document) is unclear to us, and we suspect the reviewer intended to refer to line 455 (i.e. line 502 in the track-changes document), where we state this point. |
| | Change | No change in manuscript. |

| 1.19 | RC#1 | Fig.9: it seems that the correlation patterns closely follow the border between Finland and Russia (ans to lesser degree Sweden). How can this be explained? |
|------|------|------|
| | Authors | This is an interesting observation, and we can only speculate when trying to explain the pattern. Figure 2b also shows a Finland-Russia divide in the number of fires, where more fires are found in Russia. As a consequence, the data-driven model may have been better tuned to Russian conditions as compared to Finnish conditions, whereas the FWI performance is independent of the fire occurrence density. This may be one reason for the higher correlations between the two approaches in Russia compared to eastern Finland. We have commented on this in the revised manuscript. |
| | Change | Line 663-670 |

"The varying grid-wise rank correlation between the data-driven model and each of the two FWI metrics (Fig. 9) underscores that fire danger probability maps produced by the two different approaches are different despite their similar and skilful overall performance. An interesting spatial pattern, is the notable difference in correlation closely following the Russian-Finnish border, with the higher correlations found in Russia. A likely reason for this is the fact that the data-driven model is better tuned to Russian conditions as compared to Finnish conditions due to the relatively higher number of fires in Russia (Fig. 2), whereas the FWI performance is independent of the fire occurrence density. The spatially varying correlation between FWI and the data-driven model, highlights the…"

| 1.20 | RC#1 | Can it be that the superior skill of FWI over the RF model is because FWI describes anomalous conditions whereas your model more relates to describing fire weather climatology and spatial patterns? |
|------|------|------|
| | Authors | In our understanding, FWI does not describe anomalous conditions, but rather estimates moisture (in surface, intermediate and deep organic layers) and potential for spreading regardless of what is "normal". Since soil moisture anomaly is a dominant predictor in the data-driven model, the emphasis on anomalous conditions is a more notable feature of the data-driven model rather than FWI. |
| | Change | No change in manuscript. |

| 1.21 | RC#1 | l496: In this context, reference should me made to Forkel et al., 2012, who showed that antecedent moisture conditions are better predictors of fire occurrence in a Boreal environment than FWI and precipitation anomalies. |
|------|------|------|
| | Authors | Thank you for this good suggestion, we have included it in the revised manuscript. |
| | Change | Line 545-546 (we also added the reference in line 29) |

*"Soil moisture has been found a better predictor for burned area than precipitation anomalies in another Boreal region (Baikal region; Forkel et al., 2012), supporting our findings. Studies…"*

| 1.22 | RC#1 | l498: to what extent is soil moisture an indicator of litter fuel conditions? This is usually where fires start, not in the tree crowns. |
|------|------|------|
| | Authors | We expect a strong relation between shallow soil moisture and litter fuel conditions (favourable fuel conditions for low soil moisture). We agree with your statement and have added a remark about this in the manuscript. |
| | Change | Line 550-551 |

*"Thus, soil moisture may be considered an indicator of litter fuel moisture conditions."*

| 1.23 | RC#1 | l531: This statement underestimates the role observations play in reanalysis. |
|------|------|------|
| | Authors | We agree and have made this clear in the revised manuscript. |
| | Change | Line 582-583 |

*"… despite the fact that they combine observations with modelled data."*

| 1.24 | RC#1 | l533-534: Could it be that wind is not directly but indirectly related, i.e. by the dominant weather patterns? High-pressure conditions, which are favourable to fire weather, are typically associated with low wind speeds. Vice-versa, westerlies bring high wind speeds and precipitation. |
|------|------|------|
| | Authors | Yes. We have added a comment about this in the revised manuscript. |
| | Change | 587-589 |

*"The wind related predictors may have been selected due to (…), and its indirect role through the link between wind and dominant weather patterns."*

| 1.25 | RC#1 | l539: are latitude and months of the year not already implicitly included in the other predictors? |
|------|------|------|
| | Authors | In some ways, yes, but they could have guided the model in cases such as the example presented commenting on the different effect of SPEI3 during the growing season as compared to the snow accumulation period (line 595-597). |
| | Change | No change in manuscript. |

| 1.26 | RC#1 | l545: vegetation variables like fAPAR and LAI would be more obvious candidates than NDVI as these are simulated by DGVMs (which is an argument you brought up earlier). |
|---|---|---|
| | Authors | As commented on earlier (comment 1.02), we excluded vegetation variables represented by DGVMs and limited the choice of variables to what is available from more common climate models (not including dynamic vegetation). Our reasoning for choosing NDVI is given in line 440-442. |
| | Change | No change in manuscript. |

| 1.27 | RC#1 | l546: Vegetation Optical Depth from microwave satellites has been proposed as fuel moisture indicators (e.g Forkel et al., 2017, 2019). |
|---|---|---|
| | Authors | We have adapted the text to acknowledge that remotely sensed vegetation properties has previously been found useful for predicting burned area on a global scale. |
| | Change | Line 600-601 |
| | | "Remotely sensed vegetation characteristics have previously proved useful for predicting burned area on a global scale (Kuhn-Regnier et al, 2021; Forkel et al, 2017)." |

| 1.28 | RC#1 | l582: Several studies have done this before as proved by the references below. Please rephrase. |
|---|---|---|
| | Authors | We have added "for Fennoscandian wildfire danger" in the revised manuscript. |
| | Change | Line 654-655 |
| | | "To our knowledge, our study is the first in which a data-driven model is developed for Fennoscandian wildfire danger, by means of training on transnational datasets derived from satellite imagery over multiple years at a sub-yearly time step." |

| 1.29 | RC#1 | l611: I'd be careful with the word easily here as in other regions others drivers can be dominant, some of which may not even have been originally tested here. Besides, high-quality datsets such as the EOBS and observation-heavy reanalysis data may be unavailable or have reduced skill, respectively, in other regions and hence lead to a different model. Also fire management is different in many parts of the globe (e.g. rangeland burning management in Africa or deforestation). |
|---|---|---|
| | Authors | Yes, we agree and have removed the word easily from this sentence, and in a sentence in the conclusion. |
| | Change | Line 694 and 749 |

**Response to comments by Referee #2 (RC#2)**

| 2.01 | RC#2 | This manuscript uses machine learning methods to predict fire danger in Fennoscandia at approximately 0.25 degree spatial scale for 2001-2019. Here, the authors are using official statistics compared to MODIS burned area, with predicted fire danger probability models compared to the results from the Canadian Fire Weather Index. The method is novel and the comparison is rigorous, but the data and approach need to be explained more – and at times even cited better – to assess the efficacy of the model. |
|---|---|---|
| | | In general, this manuscript needs to be revised in order to understand why this method may be useful for predicting fire danger probabilities. |
| | Authors | We have stated our motivation more clearly in the abstract. In the manuscript body, we believe the usefulness of the method is sufficiently justified. The reasoning is introduced, discussed and concluded; it links the background and the objectives (line 99-111) in the introduction, it is discussed in line 589-592 and Sect 4.5, and it is emphasised in the conclusion (line 637-640 and line 660-664). |
| | Change | Line 4-5 |
| | | "Data-driven models are suitable for identification of dominant factors of complex and partly unknown processes, and can both help improve process-based models and work as independent models." |

| 2.02 | RC#2 | First, the authors should explain what fire danger is as opposed to fire occurrence. |
|---|---|---|
| | Authors | We agree, and have clarified the difference between fire danger and fire occurrence in the introduction. |
| | Change | Line 68-70 |
| | | "Fire danger can be defined as the weather conditions that can trigger and sustain wildfires (Ranasinghe et al, 2021), and thus differs from (and is a prerequisite for) fire occurrence that additionally require an ignition." |

| 2.03 | RC#2 | Second, why are burned area data used as 'fire occurrence' when satellite-based active fire detections are available? |
|---|---|---|
| | Authors | The burned area data is used to get a binary fire/no-fire dataset based on the same resolution as found for many global climate models, to see if a data-driven model is able to make predictions of an observation-based dataset existing at this spatial scale. The active fire products detect burning at the time |

of overpass given relatively cloud-free conditions, which can be a problem for parts of Fennoscandia that are seldom cloud-free. The burned area product is considered less sensitive to cloud-cover and time of overpass. Further, by detecting the structural consequences of fires, the burned area product have a more direct relevance to climate-relevant consequences, such as albedo and ecosystem functioning. We will make a comment on this in the revised manuscript. We acknowledge that more analyses comparing different target datasets would be an interesting continuation of our study. We made one such comparison of the target dataset by including a fire record of Norway. We chose this over a satellite-based active fire detection dataset because it clearly separates each fire occurrence from others and all known occurrences are registered regardless of the (e.g. heat) signal captured by the satellite. We have added a paragraph in Sect 4.3 to discuss this.

| | |
|---|---|
| Change | Line 640-645 |

Note    "As an alternative to burned area, satellite-based active fire products can be used to construct a fire occurrence dataset. The active fire products detect burning at the time of overpass given relatively cloud-free conditions, which can be a problem for regions within Fennoscandia that are seldom cloud-free. We chose to apply the burned area product because it is considered less sensitive to cloud-cover. Further, the burned area product have a more direct relevance to climate-relevant consequences, such as albedo and ecosystem functioning. In addition, an independent target dataset was included for comparison, i.e. a local fire record of Norway."

2.04    RC#2    Finally, the manuscript does not describe fully many of the datasets used, including where to obtain them and what their uncertainty are.

Authors    In the data section, we have added details, explanations and citations when lacking. We have included information of their uncertainties (when available).

Change    Line 165-171 (burned area), line 184-189 (Norwegian fire record), line 264-268 (snow and soil moisture) and line 278-279 (fraction of burnable area).

Burned area: "The main reflectance data used are daily surface reflectance information in the red and Near Infrared bands (more details found in Pettinari et al., 2019). Data uncertainties are related to a potential underestimation of the actual burned area due to cloud cover, haze or other low quality of the observations. The fire burned area dataset is available both as a 0.25° longitude/latitude regular grid product and as a pixel product of 250 m resolution. We chose to use the grid product to investigate if a data-driven model is applicable for use at the spatial scale of the state of the art global climate models. Further, spatial dependency of fires (e.g. the same fire occurring in two or more cells) is reduced when using the coarser scale of the grid product as compared to the pixel product."

Norwegian fire record: "The dataset comprises all fires registered in grass, cultivated land, forests and uncultivated land, regardless of ignition source. The

data is based on the fire and rescue service reporting system in Norway (brann-og redningstjenestens rapporteringssystem; BRIS). There is no lower limit of burned area in this dataset, as it is based on fire responses of the fire department. The point locations in the dataset are the fire response attendance locations. Although these locations may not overlap with the locations where the fire started, we consider this uncertainty of minor importance at the 0.25° spatial grid applied in the study."

Snow and soil moisture: "As Fennoscandia covers a wide range of latitudes and altitudes, snow is still present in our dataset for some months and grid cells, although the months analysed were limited to April–October. The volumetric soil water is the volume of water in a given soil layer of the ECMWF Integrated Forecasting System, and is associated with the soil texture, soil depth, and the underlying groundwater level. The volumetric soil water in soil layer 1 (0–7cm) is one of the best performing datasets of established satellite- and model-based shallow soil moisture products (Beck et al., 2021)."

Fraction of burnable area: "This index represents the fraction of each grid cell that corresponds to vegetated land cover that could burn, i.e. excluding water bodies, permanent snow and ice, urban areas and bare areas. It is based on the Copernicus Climate Change Service (C3S) land cover classes. Details are found in Pettinari and Chuvieco (2018)."

| 2.05 | RC#2 | Finally, the results seem to indicate that a single shallow soil moisture variable is driving the predictions (which is not usually considered in fire danger modeling like FWI). A major revision and resubmission is recommended. |
| | Authors | The results indicate that a shallow soil moisture variable is the dominant predictor, however not sufficient alone to make a good prediction (emphasised e.g. in line 551-553). As you state, soil moisture is usually not considered in fire weather indices such as the FWI (this is commented on in general terms in line 722-723). We added the point made about soil moisture not considered in FWI in Sect. 4.5. |
| | Change | 688-689 |

"For example, soil moisture data is usually not considered in fire weather indices such as FWI, whereas shallow soil moisture anomaly was found the most dominant predictor by the data-driven model for Fennoscandia."

**Specific comments:**

| 2.06 | RC#2 | 1. The title is "A data-driven prediction model for Fennoscandian wildfires" but the thesis of the paper is to produce spatiotemporally resolved fire danger probability maps – which is not quite the same as predicting wildfires. Consider revising the title to be more specific. |

| | Authors | The authors agree, and have changed the title to "A data-driven model for Fennoscandian wildfire danger" |
|---|---|---|
| | Change | Changed the title. |

| 2.07 | RC#2 | 2. Line 19: "which stores approx. 30% of the world's soil carbon pool" needs a citation |
|---|---|---|
| | Authors | This is stated in the paper cited in the end of the sentence, i.e. Flannigan et al. (2009): *"Boreal regions store about 30% of the world's soil carbon pool…"* |
| | Change | No change in manuscript. |

| 2.08 | RC#2 | 3. Lines 26-27: "However, to the best of our knowledge, fire studies of the European boreal zone are limited." needs a citation. |
|---|---|---|
| | Authors | We have not found a paper stating this specifically, and the statement here is therefore based on our literature search. This is why we emphasise that it is "to the best of our knowledge" in the beginning of the sentence. |
| | Change | No change in manuscript. |

| 2.09 | RC#2 | 4. Line 144: What is the spatial resolution of a European Space Agency Climate Change Initiative (ESA145 CCI) product version 5.1.1cds? Please include that. |
|---|---|---|
| | Authors | The spatial resolution is 0.25 deg longitude/latitude. We agree that we should state this earlier in the paragraph, and have changed the text accordingly (note that the text is also changed based on comment 1.01 and 2.04). |
| | Change | Line 168-171 |
| | | "The fire burned area dataset is available both as a 0.25° longitude/latitude regular grid product and as a pixel product of 250 m resolution. We chose to use the grid product to investigate if a data-driven model is applicable for use at the spatial scale of the state of the art global climate models. Further, spatial dependency of fires (e.g. the same fire occurring in two or more cells) is reduced when using the coarser scale of the grid product as compared to the pixel product." |

| 2.10 | RC#2 | 5. Line 146-147: "and is based on Terra Moderate Resolution Imaging Spectroradiometer (MODIS) Reflection information" is not correct way to right this. It should be "the reflectance product of the Moderate Resolution Imaging Spectroradiometer (MODIS) sensor on the Terra satellite". Can the authors please specify which reflectance information is used? Daily surface reflectance? |
|---|---|---|
| | Authors | Thank you for pointing out the correct writing; this is corrected in the revised manuscript. The main source of data are daily surface reflectance information |

in the red and Near Infrared bands. The algorithm theoretical basis is found under documentation at the reference given (specifically http://datastore.copernicus-climate.eu/documents/satellite-fire-burned-area/D1.6.2-v1.0_ATBD_CDR_BA-FireCCI_MODIS_v5.1cds_PRODUCTS_v1.0.1.pdf, which is based on https://climate.esa.int/media/documents/Fire_cci_D2.1.3_ATBD-MODIS_v2.0.pdf). We have included the reference and details in the revised manuscript.

| | |
|---|---|
| Change | Line 164-166 |

"…is based on the reflectance product of the Moderate Resolution Imaging Spectroradiometer (MODIS) sensor. The main reflectance data used are daily surface reflectance information in the red and Near Infrared bands (more details found in Pettinari et al., 2019)."

| | | |
|---|---|---|
| 2.11 | RC#2 | 6. Section 2.2 Norwegian fire occurrence dataset – the authors have not provided a citation to the dataset, where it can be accessed, and how it is collected. Are these truly wildfires or are these fires from all ignition sources (lightning plus human-caused)? Is there a burned area minimum that fires must meet to be included in this wildfire dataset? Please describe this dataset more. |
| | Authors | We assume the reviewer is referring to Sect. 2.1.2 and not 2.2 here. We have provided more details and citation to the dataset in the revised manuscript. We are unsure what you mean by "truly wildfires" (do you mean only the wildfires ignited by lightning?) as opposed to "fires from all ignition sources". The dataset comprise all fires in grass, cultivated land, forests and uncultivated land, regardless of ignition source. We do not define wildfires depending on the type of ignition source in our study. The data are based on the fire and rescue service reporting system in Norway (*brann- og redningstjenestens rapporteringssystem; BRIS*). There is no lower limit of burned area in this dataset, as it is based on fire responses of the fire department. |
| | Change | Reference to the data in line 182, and details in line 184-189 |

"The dataset comprises all fires registered in grass, cultivated land, forests and uncultivated land, regardless of ignition source. The data is based on the fire and rescue service reporting system in Norway (brann- og redningstjenestens rapporteringssystem; BRIS). There is no lower limit of burned area in this dataset, as it is based on fire responses of the fire department. The point locations in the dataset are the fire response attendance locations. Although these locations may not overlap with the locations where the fire started, we consider this uncertainty of minor importance at the 0.25° spatial grid applied in the study."

| | | |
|---|---|---|
| 2.12 | RC#2 | 7. Line 164: Why were the months April – September selected? |

| | Authors | The Norwegian fire occurrence dataset must cover the same months as the satellite based fire occurrence dataset, and the reason for omitting October to March in the satellite based fire occurrence dataset is given in line 177-178 (few fire occurrences). We have clarified this in the revised manuscript. |
|---|---|---|
| | Change | Line 193-194 |
| | | "Data covering the same season and period as the satellite-based fire occurrence dataset were selected, i.e. April–September 2016–2019." |

| 2.13 | RC#2 | 8. Figure 3: The authors are using burned area from the European Space Agency Climate Change Initiative (ESA145 CCI) product version 5.1.1cds but noting it as fire occurrence and number of fires. Can the authors describe how this was done with the burned area product? |
|---|---|---|
| | Authors | The transition from burned area to fire occurrence is explained in Sect. 2.1.1 (line 172-178), and the transition from the national record to the Norwegian fire occurrence dataset is described in Sect. 2.1.2 (line 190-194). |
| | Change | No change in manuscript. |

| 2.14 | RC#2 | (8 continued.) Is this the most appropriate comparison of burned area to number of fires in the official statistics? What is the original spatial resolution and what is lost when aggregated to 0.25 degrees? |
|---|---|---|
| | Authors | None of the two datasets is directly comparable to the number of fires in official statistics because they are both aggregated in space and time. It is not an aim of the study to make the datasets directly comparable to official statistics, but rather see if a data-driven model is able to predict fire occurrences at the spatiotemporal resolution (0.25 deg regular grid and monthly time step) used in the study. The original spatial resolution of the burned area product is 250m. We have not evaluated what is lost when aggregated to 0.25 degrees, as the aggregated version is an established and verified dataset publically available. However, known uncertainties with the different fire datasets applied are commented on when introduced in the revised manuscript. |
| | Change | Line 165-171 and 184-189 (see changes made in response to comment 2.04) |

| 2.15 | RC#2 | 9. Line 230: Can the authors explain how snow cover was used? Especially since the model was limited to monthly values from April to September over the period 2001–2019. |
|---|---|---|
| | Authors | The (fractional) snow cover is a continuous variable describing the fraction of a given grid cell covered by snow at a given time step (we use the monthly averaged data), and was used as a potential predictor. Our study region cover a wide range of latitudes and altitudes, and snow cover is present in some grid |

cells and months also in the period analysed. We have added a comment about this in Sect 2.3.3.

| Change | Line 263-265 |
|---|---|

"As Fennoscandia covers a wide range of latitudes and altitudes, snow is still present in our dataset for some months and grid cells, although the months analysed were limited to April–October."

| 2.16 | RC#2 | 10. Line 235: The land cover data and fraction of burnable area is not well described. Which land covers? Why were those chosen? Are all vegetation types are included? |
|---|---|---|
| | Authors | Because the dataset is publically available, we do not elaborate on the details choices made in their creation. We have added a short description and a reference for interested readers to look up. |
| | Change | Line 278-279 |

"This index represents the fraction of each grid cell that corresponds to vegetated land cover that could burn, i.e. excluding water bodies, permanent snow and ice, urban areas and bare areas. It is based on the Copernicus Climate Change Service (C3S) land cover classes. Details are found in Pettinari and Chuvieco (2018)."

| 2.17 | RC#2 | 11. Line 241-242: Can the authors provide citations for this statement (and for Norway and Sweden, specifically): "We chose FWI because it is developed for boreal forests and because it is used for fire danger forecasts in large parts of Fennoscandia (Norway and Sweden)." |
|---|---|---|
| | Authors | We have provided citations for this statement in the revised manuscript (for Norway and Sweden, specifically). We have included 'Canadian' (i.e. "…is developed for (Canadian) boreal forests…") to clarify that it was not originally developed for Fennoscandia. |
| | Change | Line 283-284 |

"We chose FWI because it is developed for (Canadian) boreal forests and because it is used for fire danger forecasts in large parts of Fennoscandia (Norway and Sweden: Norwegian Meteorological Institute, 2022; Swedish Meteorological and Hydrological Institute, 2022)."

| 2.18 | RC#2 | 12. Figure 6: Should readers interpret Figure 6 as the only important variable to be soil moisture anomalies in the layer 7-28 cm? It would be helpful for the authors to spend more time explaining why this figure is important for creating a data-driven model, i.e., variable selection. |
|---|---|---|
| | Authors | No, Figure 6 should not be interpreted this way. The figure shows the importances of the subset of predictors used in the final data-driven model, |

and is therefore rather showing the opposite; multiple predictors are important, and a model of the soil moisture anomaly alone would not perform well. This is further emphasised by Figure S1, which shows that model performance reduces when reducing the number of predictors, and by Figure 7, which illustrates that swvl2_anomaly alone is not a sufficient predictor. See e.g. line 500-502. This figure is not important for creating a data-driven model, rather it is a result of the final data-driven model.

Change    No change in manuscript.

2.19   RC#2    13. Table 1: Should NDVI be included in this as a potential predictor?

Authors   We considered including NDVI in this table, but concluded not to because the NDVI experiments were performed separately from the main analysis.

Change    No change in manuscript.

2.20   RC#2    14. Figure 8: The red-blue scheme is not colorblind safe. Can the authors change these figures to make them colorblind safe? Tools like colorbrewer can help.

Authors   We tested the figures for colour blindness using https://www.color-blindness.com/coblis-color-blindness-simulator/ and the app "Color Blind Pal". We did not find any difficulty for the different colour blind views with this figure. Given your comment, we wonder if we have overlooked a colour blind view. If so, please let us know for which colour blind view this figure is a problem for, so we can correct it. It is of high priority to us to make the figures interpretable for all colour views.

Change    No change in manuscript.

2.21   RC#2    15. Figure 8: At first look, a reader may think that the fire danger probability maps did not perform well, especially compared to the satellite-based fire occurrence (which is really burned area dataset). Using the active fire products from MODIS or VIIRS may provide a better match than the burned area. Further, consider changing the title and better explaining fire danger in the Introduction so that interpretation of the Results is more straightforward.

Authors   The satellite-based fire occurrence dataset is used to construct the model, which is why we use this dataset in Figure 8. We are unsure if the active fire products would provide a better match, as the main aim is to map regions of fire danger probability and not to predict fire occurrences as such. Rather, the lack of no fire occurrences is partly related to the lack of ignition source given a high fire danger probability (risk of fire). However, regions with fire occurrences are often mapped with high probability, indicating a good model prediction. As for the title, see our answer to your comment 2.06. We have further clarified the term 'fire danger' to ensure it is well understood.

| | | |
|---|---|---|
| Change | Changed the title, line 69-70 (fire danger) and line 662-663 (ignition) | |
| Note | Title: "A data-driven model for Fennoscandian wildfire danger" | |
| | Fire danger: "Fire danger can be defined as the weather conditions that can trigger and sustain wildfires (Ranasinghe et al, 2021), and thus differs from (and is a prerequisite for) fire occurrence that additionally require an ignition." | |
| | Ignition: "High fire danger probabilities are also found in data points without fire occurrence. This was expected, as ignition is needed for a fire to occur." | |

| | | |
|---|---|---|
| 2.22 | RC#2 | 16. Figure 9: Same comment as for Figure 8. Is this colorblind safe? The colors chosen are hard to interpret, particularly in Figure 9c. |
| | Authors | We tested the figure for colour blindness (see our response to comment 2.20) and could not find an issue with the colours for the different colour visions. The same colour scale is used for all three maps to ease the comparison. Figure 9c shows high correlations (above 0.8 for the whole study domain) and thus, is only represented by two of the colours from the scale. |
| | Change | No change in manuscript. |

| | | |
|---|---|---|
| 2.23 | RC#2 | 17. Line 500: Most of the figures and results in the manuscript highlight the importance of swvl2_anomaly only. The manuscript needs to better describe the input and importance of other variables. |
| | Authors | We disagree that most figures and results highlight the importance of swvl2_anomaly only. There is only one figure (Fig. 7) in which swvl2_anomaly is the only predictor shown. This is justified by the relatively high importance of this predictor as compared to other predictor as shown in Fig. 6. All other figures relating to the predictors, show either all potential predictors (Fig. 4), or all selected predictors of a given model (Fig. 6, S3b, S4b, S5b and S13b). All input variables are described in Sect. 2.3, and the selected variables other than swvl2_anomaly are discussed in the lines 554-590; following the line of your comment. Commenting on the role of all predictors in more details, also those of less importance, we believe would lengthen an already long text and divert the attention from the key findings. We have therefore chosen not to do so. |
| | Change | No change in manuscript. |

| | | |
|---|---|---|
| 2.24 | RC#2 | 18. Lines 535: The authors need to better evidence to say that reanalysis products are helpful when what was used in this study is mainly reanalysis. |
| | Authors | Stating that "the use of reanalysis products is useful" does not imply that it is more useful compared to another alternative, but simply that reanalysis products can be used to construct a well-performing model. |
| | Change | No change in manuscript. |

| 2.25 | RC#2 | 19. Conclusions: Since the subsurface soil layers are the best predictors, can the authors provide some description of this dataset and the uncertainties / validation of the product? This is not described in section 2.3.3. |
|------|------|------|
| | Authors | We agree that this can be a valuable information, and have added some description and refer to an evaluations study in the revised manuscript. |
| | Change | Line 265-268 |

"The volumetric soil water is the volume of water in a given soil layer of the ECMWF Integrated Forecasting System, and is associated with the soil texture, soil depth, and the underlying groundwater level. The volumetric soil water in soil layer 1 (0–7cm) is one of the best performing datasets of established satellite- and model-based shallow soil moisture products (Beck et al., 2021)."

| 2.26 | RC#2 | 20. The authors have not shared the data or code and these should be provided. How was this study conducted? In R? In MATLAB? Please provide these details. |
|------|------|------|
| | Authors | The datasets are openly available online, except for the details concerning the Norwegian fire dataset, for which we have provided the source. We have added a 'code availability' and a 'data availability' section at the end following the Copernicus template where we repeat the information given in the data section and acknowledgements related to data availability. We support the general efforts to make code used in publications available to make analyses reproducible. Unfortunately, the code is not in in a state appropriate for sharing. However, we have added "Code is available upon reasonable request to the corresponding author" under the 'code availability section'. Here, we also state the tools used for the calculations and visualisations. |
| | Change | Line 760-772 |
| | Note | "*Code availability.* Code is available upon reasonable request to the corresponding author. Command line Climate Data Operators (CDO; Schulzweida, 2021) and the Python package xarray (Hoyer and Hamman, 2017) were used for processing the NetCDF files. SPI and SPEI calculations were performed using the SCI package in R (Gudmundsson and Stagge, 2016). The remaining calculations and visualisations were performed using Python: NumPy (Harris et al., 2020) and pandas (Pandas development team, 2020) were used for data handling, Scikit-learn (Pedregosa et al., 2011) for the construction, training and evaluation of the data-driven models, SciPy (Virtanen et al., 2020) for computation of the correlations between predictors, xskillscore (xskillscore, 2021) for calculating the correlation effective p-value between the data-driven models and FWI metrics, and Matplotlib (Hunter, 2007), cartopy (Met Office, 2010 - 2015) and seaborn (Waskom, 2021) for the visualisations of the results. |

*Data availability*. All data used are available online (urls accessed 16.06.22): E-OBS (https://doi.org/10.24381/cds.151d3ec6), ERA5-Land

(https://doi.org/10.24381/cds.151d3ec6 and
https://doi.org/10.24381/cds.68d2bb30) and v5.1.1cds
(https://doi.org/10.24381/cds.f333cf85) are available at the Copernicus
Climate Change Service (C3S) Climate Data Store, NDVI data at NASA's Land
Processes Distributed Active Archive Center (LP DAAC;
e4ftl01.cr.usgs.gov/MOLT/MOD13C2.006/), and the Norwegian wildfire record
at the services by Norwegian Directorate for Civil Protection (DSB;
http://www.brannstatistikk.no/)."

**Response to comments by Editor**

| 3.01 | Editor | (a) Please check the notification from and review file validation and address these issues in the revised version |
| --- | --- | --- |
| | Authors | In our understanding, this comment intersects with comments by referee #2 (comments 2.04, 2.14 and 2.25). We therefore refer to our changes made with regards to those. |
| | Change | See comments 2.04, 2.14 and 2.25. |

| 3.02 | Editor | (b) For all maps: Add geographical coordinates, N arrow and scale. |
| --- | --- | --- |
| | Authors | We have added lines of latitudes and longitudes in all maps (Fig. 2, 3, 8, 9, S6, S7, S8, S9, S11 and S12). Because we use a regular longitude/latitude projection, scales other that what is given by the latitude and longitude lines are not included. The north direction has a one-to-one correspondence with the coordinates, and we have therefore not included a North arrow. If you think otherwise, please let us know and we will add the arrow anyway. |
| | Change | Added lines of latitudes and longitudes in all maps (Fig. 2, 3, 8, 9, S6, S7, S8, S9, S11 and S12) |

---

## Author Response (AR2)

Dear Margreth Keiler,

Thank you for taking the time to solve the communication and technical submission issues, as well as carefully reading the manuscript again, in order to proceed with the review process of our manuscript. Hereby, we would like to provide our point-by-point reply to the comments of Referee #1. As you know, Referee #2 recommended to accept the manuscript as is, and had no further comments.

Original Referee #1 comments are marked by 'RC#1', our responses by 'Authors', and corresponding changes to the manuscript by 'Change' (line numbers refer to the track-changes file). All text changes are visible in the track-changes version of the manuscript.

Kind regards,

Sigrid J. Bakke, Niko Wanders, Karin van der Wiel and Lena M. Tallaksen

**Response to comments by Referee #1 (RC#1)**

| 1.01 | RC#1 | Introduction - The Flannigan et al. (2009) does not find that boreal regions store 30% of the world's soil carbon pool. Within that paper, they are citing someone else's work "...but store an estimated 30% of the world's soil C pool (Gorham, 1991)." Is this still the estimate? |
|---|---|---|
| | Authors | Thank you for this notice; we have replaced the percentage from Gorham (1991) with information based on an updated estimate of the boreal carbon stock from Bradshaw and Warkentin (2015) in our revised manuscript. |
| | Change | Line 21-23 |

"In the boreal region, which *have the largest carbon stock of all major global forest biomes*, fires are the major stand-renewing agent and play a major role in carbon storage and emissions (*Bradshaw and Warkentin, 2015;* Flannigan et al., 2009)."

| 1.02 | RC#1 | Section 2 - The authors must consider that the spatial resolution of burned area data, which is nominally 250 m, means that it will miss small fires. Active fire products, at 1 km or 375 m, are capable of detecting fires 1/10 the size the pixel resolution [see Patricia Oliva and Wilfrid Schroeder, "Assessment of VIIRS 375 m active fire detection product for direct burned area mapping", Remote Sensing of Environment, vol. 160 (2015), pp. 144–155; Tianran Zhang et al. "Approaches for synergistically exploiting VIIRS I-and M-Band data in regional active fire detection and FRP assessment: A demonstration with respect to agricultural residue burning in Eastern China", Remote Sensing of |
|---|---|---|

Environment, vol. 198 (2017), pp.407–424.] So excluding active fire products excluded smaller fires.

| | |
|---|---|
| Authors | Thank you for suggesting these two papers that emphasise active fire products' capability of detecting smaller fires as compared to burned area products. We agree that there is a lack of small fires in the burned area product used in the paper. In the manuscript, we acknowledge that the burned area data lack small fires in the abstract (line 14-15), introduction (line 57-59) and discussion (Sect 4.3).

The benefits and drawbacks of different data sets used for fire occurrences are discussed in Sect. 4.3. Here, the reasons for not selecting active fire products are provided (line 632-635). Based on your comment, we have added a remark about the capability of active fire products to detect small fires in Sect. 4.3 in the revised manuscript. |
| Change | Line 629-632 |

*"Active fire products are capable of detecting smaller fires compared to standard burned area products (Wooster et al., 2021; Oliva and Schroeder, 2015). Whereas small fire detection is improved in many regions by using active fire products, detection errors (i.e. false fires) are a problem in some regions and seasons (Wooster et al., 2021; Zhang et al., 2018)."*

| | | |
|---|---|---|
| 1.03 | RC#1 | Aggregating 250 m to 0.25° is also potentially averaging out smaller fires. This should be noted in the discussion of uncertainty, especially given that the official on-the-ground fire occurrence dataset registers all fires regardless of size. |
| | Authors | The burned areas are summed in the aggregation from 250m pixels to 0.25° grid cells. Consequently, fire occurrence for a given grid cell at a given month is set to one or true for all grid cells with at least one underlying pixel with burned area >0. This reduces the number of fires in cases where more than one underlying pixel had burned area >0 *regardless of the size* of the detected fires (burned areas). The same reduction happens when we transform the point based national record to a gridded data set of Norwegian fire occurrences (ref. line 184-187). Thus, the difference between the satellite based and on the ground based fire occurrence data set is mainly stemming from the inability of the satellite product to detect small fires, and not the spatial aggregation. We argue for our choice of the *gridded* burned area data in Sect 2.1.1 (line 163-165): the gridded data set matches the spatial scale of the state of the art climate models and it reduces the risk of spatial dependency between fire occurrences (i.e. same fire occurring in two or more cells). The problem of detecting small fires in the burned area product as compared to the national record is discussed in Sect 4.3. |
| | Change | No change in manuscript. |

| 1.04 | RC#1 | Section 2.1.1 – How many fires are included in Figure 2? This is should be reported in the text and ideally broken down by country. |
| | Authors | We edited the text in Sect 2.1.1 to include the number of fire occurrences in the revised manuscript. |
| | Change | Line 174 |

"There is an extreme imbalance between the two classes (fire and no-fire) in the target variable, with only *1439 of the 444030 data points (*0.3%*)* classified as fire."

| 1.05 | RC#1 | Figure 3 – What is the spatial resolution of Figure 3? 0.25°? Were the the Norweigan fire occurrence data aggregated to 0.25°? Or was a 0.25° grid overlaid on the Norweigan dataset? Explaining this a bit more explains why there is so much fire in Figure 3B and basically no fire in Figure 3A. |
| | Authors | The spatial resolution of Fig. 3 is 0.25°. We now provide this information in the figure caption in the revised manuscript. |
| | | The transformation of the point-based national record of historical wildfires in Norway to the 0.25° resolution is described in Section 2.1.2 (line 184-187). The reason for the large differences between Fig. 3a and Fig. 3b is mainly the lack of small fires in the satellite-based fire occurrence data, as mentioned in Sect. 4.3 (line 613-616). We have added this information in Sect 2.1.2 in the revised manuscript. |
| | Change | Figure caption of Fig. 3 (page 8) and line 192 |

"… spatial distribution over Norway (map; *0.25° resolution*), and the…"

"There are substantial differences between the two datasets*, mainly arising from the lack of small fires in the satellite-based fire occurrence dataset*."

| 1.06 | RC#1 | Section 4.5 – What exactly is the value-added for a data-driven model to predict fire danger probability for actionable management of wildfires? In Figure 8, the months of May, June, July have more detail in the Model Prediction for high fire danger than FWI, but a very similar pattern. Additionally, explain to the reader how this developed could be used for fire season management if using near-real-time data and/or near term climate model outputs. This is not obvious in this section. |
| | Authors | The value added (as discussed in Sect. 4.5) include improved trust (and knowledge about the uncertainties) in the fire danger maps, insight into which environmental indices one should consider when improving process-based models, and the transferability of this method to other regions. For example, high agreement between the two approaches gives improved trust in the fire danger for the given region and month. The similarities between the model prediction and FWI varies over time. As you mention, they are similar in May-July 2018, whereas in April, August and September 2018 they |

are more diverse (Fig. 8). Figure 9 summarises the similarities/differences between the approaches for the full test set.

As for your second comment (starting with "Additionally.."), we assume you refer to our comment that we regard data-driven models as valuable contributions to fire forecasting (line 701-702). Here, we refer to data-driven models in general, and not our model specifically, which was not developed for forecasting.

| | |
|---|---|
| Change | No change in manuscript. |

| | | |
|---|---|---|
| 1.07 | RC#1 | Data availability - The authors need to better describe how to access the Norwegian wildfire record at https://www.brannstatistikk.no/. Mention that the site is in Norwegian. How would someone request the data from here? Do you have to be a resident of Norway or a citizen of Norway to request and/or access the data? Is the data allowed to be shared or posted publicly? This is an important data set for the findings of this analysis but how someone would replicate this study by accessing this data is unclear. |
| | Authors | We agree it is valuable to make the information about data access as clear as possible, and we have added information based on your suggestions in the revised manuscript. |
| | Change | Line 754-755 |

*"Note that the DSB webpage is in Norwegian. Data are freely available, and in case of any questions regarding the data, please use the contact information provided by the webpage."*

| | | |
|---|---|---|
| 1.08 | RC#1 | References - The following reference is now out of date and the brokered links do not appear anymore at https://climate.esa.int/en/projects/fire/key-documents/: |
| | | Pettinari, M., Lizundia-Loiola, J., and Chuvieco, E.: Algorithm Theoretical Basis Document: CDR Fire Burned Area (brokered from CCI Fire Burned Area), available at: http://datastore.copernicus-climate.eu/documents/satellite-fire-burned-area/D1.6.2-v1.0_ATBD_CDR_BA-FireCCI_MODIS_v5.1cds_PRODUCTS_v1.0.1.pdf, 2019. |
| | Authors | We use the same reference as the Copernicus Data Store (CDS), from which we downloaded the burned area product (https://cds.climate.copernicus.eu/cdsapp#!/dataset/satellite-fire-burned-area?tab=doc). This document builds on other documents available at ESA's webpage. Because we used data from CDS, we find it more appropriate to use the connected CDS reference. The link we provide in the reference is working, and the reference is to our knowledge not out of date (it is still used by CDS). |
| | Change | No change in manuscript. |